# Learning What to Fail On: Failure-Mode Contextual Bandits for Adversarial Data Curation

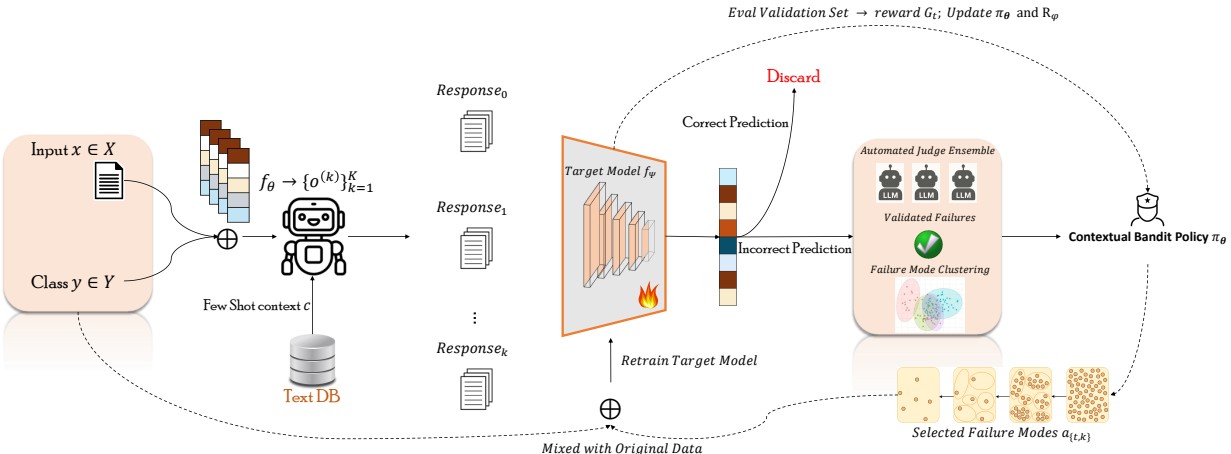

Figure 1: Overview of the proposed failure-mode contextual bandit curation framework. Given an input-label pair, retrieval-augmented prompting generates candidate outputs. The target model filters candidates by retaining only incorrect predictions, which are then automatically validated by an LLM judge ensemble and clustered into recurring failure modes. A contextual-bandit policy selects failure modes under an adversarial budget, and selected examples are mixed with original data to retrain the target model. Validation feedback provides reward $G_t$ for updating the policy $\pi_\theta$ and critic $R_\phi$, enabling adaptive selection of high-impact failure modes across rounds.

## Abstract

We introduce a failure-aware adversarial retrieval-augmented framework for improving robustness in natural language understanding. Rather than selecting synthetic examples with a fixed reward threshold, our method formulates adversarial data curation as a failure-mode contextual bandit problem. Candidate examples are generated with retrieval-augmented prompting, filtered by the current target model, automatically validated by an LLM judge ensemble, and clustered into recurring failure modes. A stochastic policy then selects which failure modes to sample for retraining, and is updated using validation-based reward that balances robustness gains, forgetting, and data cost. This makes the data curator itself the learning agent, enabling adaptive selection of the most useful model failures across training rounds. On standard benchmarks, our approach improves RoBERTa-base accuracy from 88.48% to 92.60% on SNLI, from 75.04% to 80.95% on ANLI, and from 54.67% to 71.99% on MultiNLI, while consistently outperforming prior adversarial augmentation methods. We further demonstrate transfer to FEVER fact verification, achieving up to 79.86% FEVER score and 82.45% accuracy with RoBERTa-large. Finally, we provide a theoretical interpretation showing that, under stated assumptions, failure-mode sampling can reduce shortcut-aligned gradient contributions while inducing bounded distributional drift. By combining retrieval, automated validation, contextual-bandit failure selection, and controlled adversar-

ial retraining, our framework enables scalable robustness improvement without additional human annotation.

# 1 Introduction

Natural Language Inference (NLI), the task of determining whether a hypothesis is entailed by, contradicted by, or neutral with respect to a given premise, is a central component of many natural language understanding problems, including question answering, summarization, dialogue systems, and fact verification. More broadly, many supervised NLP tasks suffer from similar robustness limitations when confronted with adversarial or out-of-domain examples. Despite rapid progress, even state-of-the-art models remain brittle, often relying on spurious lexical cues or failing under simple syntactic and semantic variations (Glockner et al., 2018; Carmona et al., 2018). Benchmarks such as SNLI and MultiNLI (Bowman et al., 2015; Williams et al., 2018), as well as adversarial datasets (Nie et al., 2019), have driven robustness improvements but incur high annotation costs and still leave many failure modes uncovered. More recently, large-scale synthetic datasets such as GNLI (Hosseini et al., 2024) have been generated automatically, but their largely untargeted nature can dilute the adversarial patterns most useful for improving a particular target model. Existing adversarial data generation pipelines typically rely on static filtering, heuristic selection rules, or one-shot validation. As a result, they do not explicitly learn which types of failures should be prioritized as the target model evolves. This limitation is especially important because model failures are not equally useful: some expose persistent decision shortcuts, while others are noisy, redundant, or too easy to produce meaningful robustness gains. Motivated by this gap, we formulate adversarial data curation as a *failure-mode contextual bandit* problem. The learning agent is not the target classifier itself, but the data curator that decides which types of validated model failures should be sampled for retraining. Our framework first retrieves label-balanced few-shot contexts using both semantic embeddings (BGE M3 (Chen et al., 2024)) and lexical matching (BM25 (Robertson & Zaragoza, 2009)). These contexts are assembled into LLM prompts to generate challenging candidate hypotheses. Each candidate is evaluated by the current target model, and only examples that induce misclassification are passed to an automated LLM judge ensemble for label validation. The validated failures are then embedded and clustered into recurring failure modes, such as lexical shortcut failures, negation errors, entity mismatch errors, numerical reasoning failures, or contradiction confusion. A stochastic contextual-bandit policy then observes a state vector for each failure mode, including cluster size, target-model loss, uncertainty, classification margin, label distribution, retrieval score, judge agreement, novelty, and previous reward statistics. The policy selects which failure modes to sample under a fixed adversarial budget. After retraining the target model on a controlled mixture of original and selected adversarial examples, the policy receives a validation-based reward that balances robustness improvement, forgetting on the clean distribution, and data cost. A lightweight critic estimates the expected utility of each failure mode, but selection is governed by the learned policy rather than a fixed reward threshold. This design provides an explicit policy, action space, reward signal, and policy-optimization procedure for adaptive adversarial data curation. In human-free adversarial fine-tuning and transfer evaluations on NLI benchmarks, our approach improves RoBERTa-base accuracy from 88.48% to 92.60% on SNLI, from 75.04% to 80.95% on ANLI, and from 54.67% to 71.99% on MultiNLI, while consistently outperforming prior adversarial augmentation methods. Beyond NLI, we further demonstrate transfer to the FEVER fact verification benchmark. Using RoBERTa-large, our method achieves up to 79.86% FEVER score and 82.45% label accuracy, outperforming strong retrieval-augmented and synthetic-data baselines. These results indicate that failure-mode bandit curation can improve robustness across task formulations and supervision regimes while using automatically generated and automatically validated data. **Our contributions are:**

**Framework.** We propose a failure-mode contextual bandit framework for adversarial data curation. The framework requires no additional human annotation and learns which validated model-failure modes should be sampled for retraining.

**Methodology.** We introduce an adaptive curation pipeline that combines label-balanced retrieval, LLM-based candidate generation, target-model failure filtering, automated judge validation, unsupervised failure-mode clustering, and contextual-bandit selection under an adversarial data budget.

**Empirical Evaluation.** We provide empirical evidence that failure-mode bandit curation improves robustness and data efficiency compared with static adversarial augmentation, reward-threshold filtering, retrieval-only selection, and untargeted synthetic-data baselines.

**Theoretical Interpretation.** We provide an analytical interpretation showing that, under stated assumptions, failure-mode sampling can reduce shortcut-aligned gradient contributions while preserving core-feature contributions. We further show that mixture-based updates induce bounded distributional drift and that bounded utility noise causes bounded distortion in the induced sampling policy.

## 2 Background and Related Work

Improving the robustness of NLI models remains a central challenge in natural language understanding (Glockner et al., 2018; Carmona et al., 2018). Benchmarks such as SNLI (Bowman et al., 2015) and MultiNLI (Williams et al., 2018) have enabled large-scale supervised training, while ANLI (Nie et al., 2019) introduced a human-and-model-in-the-loop protocol for collecting harder adversarial examples. However, these datasets require substantial annotation effort and still leave many systematic failure modes uncovered. More recently, automated and synthetic data-generation approaches have reduced the need for human annotation. For example, GNLI (Hosseini et al., 2024) shows that large-scale generated NLI data can rival human-curated data, and Kazoom et al. (2025) proposed a training-free retrieval-augmented framework for adversarial detection and filtering. Related work on counterfactual and paraphrase generation has also been used to enrich training distributions (Li et al., 2023; Klemen & Robnik-Šikonja, 2021). Despite these advances, most existing approaches rely on static generation, heuristic filtering, or one-shot validation, and therefore do not explicitly learn which types of model failures should be prioritized as the target model evolves.

**Adversarial and Synthetic Example Generation.** Automated adversarial pipelines aim to expose and correct model weaknesses without manual curation. Minervini & Riedel (2018) generate logical-constraint-violating examples, improving robustness on SNLI and MultiNLI. Nie et al. (2020) use a model-in-the-loop setup to surface challenging examples and improve out-of-domain transfer. Iyyer et al. (2018) introduce syntactically controlled transformations for paraphrase-based attacks. Recent LLM-based pipelines further demonstrate that generated hypotheses, especially when combined with automated validation, can provide useful adversarial supervision. However, these methods typically select examples using fixed rules, confidence thresholds, heuristic filters, or limited human feedback. In contrast, our work treats adversarial data curation as an adaptive learning problem: the system identifies recurring target-model failures, clusters them into failure modes, and learns which modes are most useful for retraining.

**Retrieval for Few-Shot Prompting.** Few-shot retrieval is important for reliable LLM-based generation because the retrieved context controls both the label distribution and the semantic structure of generated examples. Dense retrieval models such as BGE (Chen et al., 2024) provide semantic similarity, while lexical methods such as BM25 (Robertson & Zaragoza, 2009) capture surface-level overlap and exact lexical cues. Hybrid retrieval can therefore provide complementary context for generating diverse and label-consistent hypotheses. In our framework, retrieval is not the main contribution by itself. Instead, it serves as the first stage of a larger closed-loop curation pipeline: retrieved examples guide generation, generated candidates expose target-model failures, and the downstream bandit policy decides which validated failure modes should be sampled for retraining.

**Reinforcement Learning, Bandits, and Data Selection.** Learning to select training examples has been studied in reinforcement-learning and curriculum-learning settings. Fan et al. (2017) propose a Neural Data Filter that learns to select useful training samples. Reinforcement-guided curricula have also been explored in structured prediction tasks such as neural machine translation, where policies learn to sequence or weight examples for improved training (Zhao et al., 2020). More recent work formulates data selection for model finetuning as a sequential decision problem, where an agent chooses subsets of data to optimize validation rewards (Jha et al., 2025). Related methods such as LearnAlign (Li et al., 2025) and RL-Selector (Yang et al., 2025) use reward feedback to select informative examples or reduce redundancy. Our method differs from these approaches in both the unit of selection and the source of supervision. Rather than selecting arbitrary training examples, we first construct a validated pool of adversarial failures induced by the current target model. These failures are then clustered into semantic failure modes, and a contextual-bandit policy

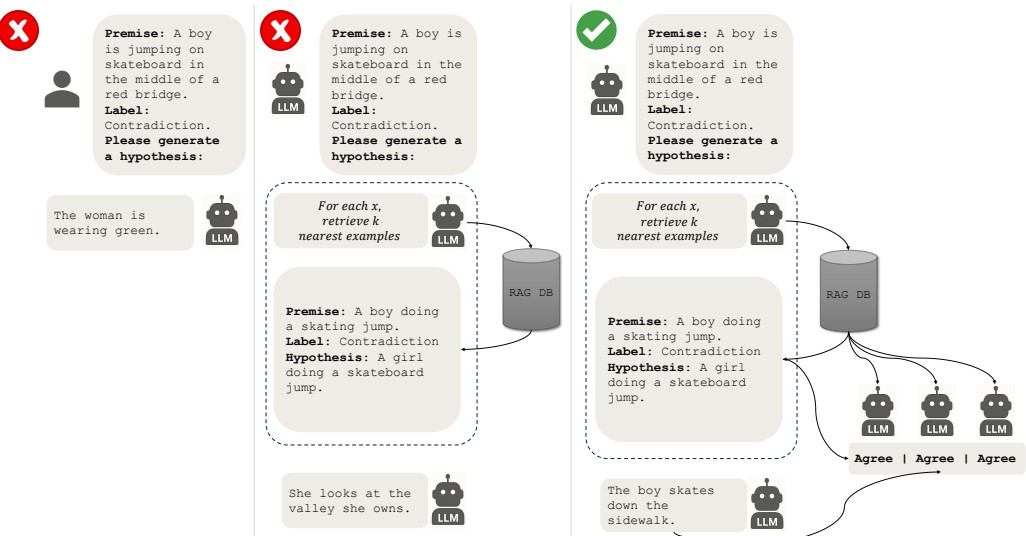

Figure 2: Example of the progressive construction of our failure-aware policy. From left to right: using only an LLM for hypothesis generation without retrieval or feedback; adding few-shot retrieval to condition generation on similar failures; and the full reinforcement-guided policy, which combines retrieval, multi-model validation, and reward-based selection to identify and reinject high-impact adversarial updates.

selects which modes to sample under a fixed adversarial budget. The reward is computed after retraining, using validation improvement, forgetting penalty, and data cost. This provides an explicit state, action, reward, and policy update while avoiding the instability and expense of per-example utility estimation.

## 3 Methodology

Let $\mathcal{D} = \{(p_i, y_i)\}_{i=1}^{N}$ denote the NLI training set, where each premise $p_i$ is paired with a label $y_i \in \mathcal{Y}$ and $\mathcal{Y} = \{\text{entail}, \text{neutral}, \text{contradict}\}$. We denote by $M^{(t)}$ the target model after $t$ rounds of failure-aware adversarial data curation, with $M^{(0)}$ trained on $\mathcal{D}$. The goal is to construct a compact adversarial training set that improves robustness without relying on large volumes of untargeted synthetic examples. We formulate this process as *failure-mode contextual bandit curation*. At each iteration, the system retrieves few-shot contexts, generates candidate adversarial examples, filters candidates that induce errors in the current target model, and validates them using an automated judge ensemble. The validated failures are then grouped into semantic failure modes. Instead of selecting individual examples by a fixed reward threshold, a trainable stochastic policy $\pi_\theta$ observes each failure mode and decides which modes should be sampled for retraining. After the target model is updated, the policy receives a validation-based reward and is optimized using a policy-gradient objective. Thus, the learning agent is the data curator, whose role is to learn which types of model failures are most useful for improving future robustness. This formulation makes the reinforcement-learning component explicit. The state is a vector of statistics describing a failure mode, the action is whether to sample from that failure mode, the reward is the downstream validation gain after retraining, and the policy is updated to maximize expected validation improvement while penalizing forgetting and computational cost. Each iteration consists of six steps: label-balanced retrieval, LLM-based candidate generation, failure filtering using the current target model, automated validation, clustering validated candidates into failure modes, and contextual-bandit selection of failure modes for retraining.

**Retrieval.** For each premise $p$, we construct a label-balanced few-shot context: $\mathcal{C}_p = \bigcup_{y' \in \mathcal{Y}} \mathcal{C}_{p,y'}$, where $\mathcal{C}_{p,y'}$ contains $k$ examples with label $y'$. Let $\mathcal{D}_{y'}$ denote the subset of training examples with label $y'$. Label-balanced retrieval prevents the prompt from being dominated by a single class and provides the generator with controlled examples from all NLI relations.

**Semantic Retrieval.** Let $E_{\text{emb}}$ denote the embedding model. Each premise $x$ is embedded as: $e_x = E_{\text{emb}}(x) \in \mathbb{R}^d$. For a query premise $p$, we compute: $e_p = E_{\text{emb}}(p)$. For each label $y'$, semantic neighbors are selected by: $\mathcal{C}_{p,y'}^{\text{sem}} = \arg\max_{\substack{S \subseteq \mathcal{D}_{y'} \\ |S|=k}} \sum_{x \in S} \cos(e_p, e_x)$. This corresponds to selecting the top-$k$ nearest neighbors in embedding space within each label group.

**Lexical Retrieval.** We index all premises using BM25 with parameters $(k_1 = 1.5, b = 0.75)$ and define the lexical relevance score: $s_{\text{BM25}}(p, x) = \sum_{w \in p} \text{IDF}(w) \cdot \frac{\text{tf}(w,x)(k_1+1)}{\text{tf}(w,x)+k_1\left(1-b+b\frac{|x|}{\text{avgdl}}\right)}$. For each label $y'$, lexical neighbors are selected as: $\mathcal{C}_{p,y'}^{\text{lex}} = \arg\max_{\substack{S \subseteq \mathcal{D}_{y'} \\ |S|=k}} \sum_{x \in S} s_{\text{BM25}}(p, x)$.

**Hybrid Retrieval.** To integrate semantic and lexical signals, for each candidate $x$ and query $p$, we compute normalized scores: $\tilde{s}_{\text{sem}}(p, x) = \frac{\cos\left(E_{\text{emb}}(p), E_{\text{emb}}(x)\right) - \mu_{\text{sem}}}{\sigma_{\text{sem}}}$, and: $\tilde{s}_{\text{lex}}(p, x) = \frac{s_{\text{BM25}}(p,x) - \mu_{\text{lex}}}{\sigma_{\text{lex}}}$. The hybrid relevance score is defined as: $s_{\text{comb}}(p, x) = \alpha\tilde{s}_{\text{sem}}(p, x) + (1 - \alpha)\tilde{s}_{\text{lex}}(p, x)$, where $\alpha \in [0, 1]$ controls the interpolation between semantic and lexical similarity. For each label $y'$, we select: $\mathcal{C}_{p,y'}^{\text{comb}} = \arg\max_{\substack{S \subseteq \mathcal{D}_{y'} \\ |S|=k}} \sum_{x \in S} s_{\text{comb}}(p, x)$. The final context is: $\mathcal{C}_p = \bigcup_{y' \in \mathcal{Y}} \mathcal{C}_{p,y'}^m$, where $m \in \{\text{sem}, \text{lex}, \text{comb}\}$ denotes the retrieval mode. The complete retrieval is summarized in Algorithm 1.

---

**Algorithm 1** Balanced Few-Shot Context Retrieval

**Input**: Premise $p$, label-partitioned dataset $\{\mathcal{D}_y\}_{y \in \mathcal{Y}}$
**Parameter**: examples per label $k$, retrieval mode $m \in \{\text{sem}, \text{lex}, \text{comb}\}$
**Output**: Few-shot context $\mathcal{C}_p$

1: $\mathcal{C}_p \leftarrow \emptyset$
2: Compute retrieval scores $s_m(p, x)$ for all $x \in \mathcal{D}$
3: **for** each label $y' \in \mathcal{Y}$ **do**
4: $\quad \mathcal{C}_{p,y'} \leftarrow \arg\max_{x \in \mathcal{D}_{y'}}^k s_m(p, x)$
5: $\quad \mathcal{C}_p \leftarrow \mathcal{C}_p \cup \mathcal{C}_{p,y'}$
6: **end for**
7: **return** $\mathcal{C}_p$

---

**Task-Specific Candidate Generation.** Given an input $x$, its label $y$, and the retrieved context $\mathcal{C}_x$, we employ a large language model to sample task-specific candidate outputs from:

$$o \sim P_{\text{LLM}}(o \mid x, \mathcal{C}_x, y). \tag{1}$$

This stochastic generation process produces a candidate set $\mathcal{O}_x^{(t)}$ for each input at iteration $t$, where $o$ denotes a task-dependent output, such as a hypothesis for NLI or an evidence claim for fact verification.

**Failure-Based Filtering.** Each generated candidate $o \in \mathcal{O}_x^{(t)}$ is evaluated by the current target model $M^{(t)}$. Let the predicted label be:

$$\hat{y}_o = \arg\max_{y' \in \mathcal{Y}} M^{(t)}(y' \mid x, o). \tag{2}$$

Candidates that are correctly classified are discarded, and only misclassified instances are retained:

$$\mathcal{O}_x^{\text{fail}} = \left\{o \in \mathcal{O}_x^{(t)} \mid \hat{y}_o \neq y\right\}. \tag{3}$$

This step focuses the remaining pipeline on examples that expose the current model's weaknesses.

**Automated Validation.** Let the candidate adversarial triples be:

$$\mathcal{Q}^{(t)} = \left\{(x, o, y) \mid o \in \mathcal{O}_x^{\text{fail}}\right\}. \tag{4}$$

Each triple is evaluated by an ensemble of automated judge models. Let the predicted label of judge $j$ be:

$$v_j(x, o) = M_j(x, o), \qquad j \in \{1, 2, 3\}. \tag{5}$$

A candidate is retained only if all judges agree with the original label: $\sum_{j=1}^{3} \mathbb{I}\left[v_j(x,o) = y\right] = 3$. The validated failure pool is therefore: $\mathcal{V}^{(t)} = \left\{ (x,o,y) \in \mathcal{Q}^{(t)} \mid \sum_{j=1}^{3} \mathbb{I}\left[v_j(x,o) = y\right] = 3 \right\}$. This unanimity constraint reduces label noise and prevents the policy from learning from corrupted reward signals.

**Failure-Mode Construction.** Rather than selecting individual failures independently, we group validated failures into failure modes. For each validated triple $q_i = (x_i, o_i, y_i) \in \mathcal{V}^{(t)}$, we compute an embedding:

$$g_i = E_{\text{fail}}\left([x_i; o_i; y_i]\right), \tag{6}$$

where $E_{\text{fail}}$ may be the same text embedding model used for retrieval or a separate sentence encoder. We then cluster the validated failure pool:

$$\left\{\mathcal{F}_1^{(t)}, \ldots, \mathcal{F}_{K_t}^{(t)}\right\} = \text{Cluster}\left(\{g_i\}_{q_i \in \mathcal{V}^{(t)}}\right). \tag{7}$$

Each cluster $\mathcal{F}_k^{(t)}$ represents a failure mode, such as lexical shortcut failures, negation errors, entity mismatch errors, numerical reasoning failures, or contradiction confusion. The clustering is unsupervised and does not require human failure labels.

**Bandit State.** For each failure mode $\mathcal{F}_k^{(t)}$, we construct a state vector $z_{t,k}$ containing normalized statistics of that cluster:

$$z_{t,k} = \left[\log(|\mathcal{F}_k^{(t)}| + 1), \bar{\ell}_{t,k}, \bar{H}_{t,k}, \bar{\mu}_{t,k}, \text{hist}_{t,k}(y), \bar{s}_{t,k}^{\text{retr}}, \bar{a}_{t,k}^{\text{judge}}, \nu_{t,k}, \bar{G}_{t-1,k}\right]. \tag{8}$$

Here, $\bar{\ell}_{t,k}$ is the mean target-model loss on the cluster, $\bar{H}_{t,k}$ is the mean predictive entropy, $\bar{\mu}_{t,k}$ is the mean classification margin, $\text{hist}_{t,k}(y)$ is the label distribution, $\bar{s}_{t,k}^{\text{retr}}$ is the average retrieval score, $\bar{a}_{t,k}^{\text{judge}}$ is the average judge agreement, $\nu_{t,k}$ is a novelty score measuring distance from previously selected failure modes, and $\bar{G}_{t-1,k}$ is the previous moving-average reward associated with similar clusters. This state summarizes both the difficulty and the diversity of each failure mode.

**Contextual-Bandit Policy.** The policy $\pi_\theta$ is a trainable stochastic selector over failure modes. For each cluster state $z_{t,k}$, the policy outputs a Bernoulli distribution:

$$\pi_\theta(a_{t,k} = 1 \mid z_{t,k}) = \sigma\left(f_\theta(z_{t,k})\right), \tag{9}$$

where $a_{t,k} \in \{0,1\}$ is the action for cluster $k$, $a_{t,k} = 1$ means selecting that failure mode for retraining, and $f_\theta$ is a small neural network. The complete action at iteration $t$ is:

$$a_t = (a_{t,1}, \ldots, a_{t,K_t}). \tag{10}$$

To encourage exploration, actions are sampled from $\pi_\theta$ during training rather than selected deterministically. At evaluation time, the policy may use greedy selection according to the learned probabilities.

Given an adversarial budget $B_{\text{adv}}$, the selected adversarial set is:

$$\mathcal{D}_{\text{sel}}^{(t)} = \bigcup_{k:a_{t,k}=1} \text{Sample}\left(\mathcal{F}_k^{(t)}, n_{t,k}\right), \tag{11}$$

where:

$$n_{t,k} = \left\lfloor B_{\text{adv}} \frac{a_{t,k}|\mathcal{F}_k^{(t)}|}{\sum_{\ell=1}^{K_t} a_{t,\ell}|\mathcal{F}_\ell^{(t)}|} \right\rfloor. \tag{12}$$

If no cluster is selected, the cluster with the highest policy probability is selected as a fallback. This avoids empty updates.

**Target Model Retraining.** The target model is updated using supervised learning on a mixture of original and selected adversarial examples:

$$M^{(t+1)} \leftarrow \text{Train}\left(M^{(t)}, \mathcal{D}_{\text{mix}}^{(t)}\right). \tag{13}$$

The construction of $\mathcal{D}_{\text{mix}}^{(t)}$ is described in Section 3.2. This update changes the environment observed by the curator, since future failure pools depend on the updated target model $M^{(t+1)}$.

**Validation-Based Reward.** After retraining, the policy receives a scalar reward based only on validation performance. Let $\mathcal{D}_{\text{val}}^{\text{rob}}$ denote the robustness validation set and $\mathcal{D}_{\text{val}}^{\text{clean}}$ denote the clean validation set. We define:

$$G_t = \Delta_{\text{rob}}^{(t)} - \beta_f \Delta_{\text{forget}}^{(t)} - \beta_c \Delta_{\text{cost}}^{(t)}. \tag{14}$$

The robustness gain is:

$$\Delta_{\text{rob}}^{(t)} = \text{Perf}\left(M^{(t+1)}, \mathcal{D}_{\text{val}}^{\text{rob}}\right) - \text{Perf}\left(M^{(t)}, \mathcal{D}_{\text{val}}^{\text{rob}}\right). \tag{15}$$

The forgetting penalty is:

$$\Delta_{\text{forget}}^{(t)} = \max\left(0, \text{Perf}\left(M^{(t)}, \mathcal{D}_{\text{val}}^{\text{clean}}\right) - \text{Perf}\left(M^{(t+1)}, \mathcal{D}_{\text{val}}^{\text{clean}}\right)\right). \tag{16}$$

The cost penalty is:

$$\Delta_{\text{cost}}^{(t)} = \frac{|\mathcal{D}_{\text{sel}}^{(t)}|}{B_{\text{adv}}}. \tag{17}$$

The first term rewards robustness gains, the second penalizes degradation on the original distribution, and the third penalizes excessive adversarial data usage. The coefficients $\beta_f$ and $\beta_c$ control the trade-off between robustness, retention, and efficiency.

**Policy Optimization.** The curator policy is optimized to maximize expected validation reward:

$$J(\theta) = \mathbb{E}_{a_t \sim \pi_\theta}\left[G_t\right]. \tag{18}$$

We update $\pi_\theta$ with a Reinforce-style policy-gradient objective:

$$\mathcal{L}_\pi(\theta) = -(G_t - b_t)\sum_{k=1}^{K_t} \log \pi_\theta(a_{t,k} \mid z_{t,k}) - \beta_H \sum_{k=1}^{K_t} \mathcal{H}\left(\pi_\theta(\cdot \mid z_{t,k})\right), \tag{19}$$

where $b_t$ is a moving-average baseline and $\mathcal{H}(\cdot)$ is an entropy regularizer that encourages exploration. The baseline is updated as:

$$b_t = \rho b_{t-1} + (1 - \rho)G_t. \tag{20}$$

We also maintain a critic $R_\phi$ that predicts the expected return of a failure mode:

$$R_\phi(z_{t,k}) \approx \mathbb{E}\left[G_t \mid z_{t,k}\right]. \tag{21}$$

The critic is trained by minimizing:

$$\mathcal{L}_R(\phi) = \sum_{k:a_{t,k}=1} \left(R_\phi(z_{t,k}) - G_t\right)^2. \tag{22}$$

Unlike a threshold-based filtering rule, $R_\phi$ is not used as a direct keep-or-discard mechanism. Instead, it serves as a learned utility estimator and variance-reduction signal for policy learning. The actual data-selection behavior is governed by the trainable policy $\pi_\theta$. The complete procedure is summarized in Algorithm 2.

---

**Algorithm 2** Failure-Mode Contextual Bandit Curation

---

**Input**: Training set $\mathcal{D}$, validation sets $\mathcal{D}_{\text{val}}^{\text{rob}}$, $\mathcal{D}_{\text{val}}^{\text{clean}}$, iterations $T$, retrieval mode $m$, budget $B_{\text{adv}}$
**Output**: Enhanced model $M^{(T)}$

1: Train $M^{(0)}$ on $\mathcal{D}$
2: Initialize policy $\pi_\theta$, critic $R_\phi$, and baseline $b_0$
3: **for** $t = 0$ to $T - 1$ **do**
4:    $\mathcal{V}^{(t)} \leftarrow \text{GENERATE\_AND\_VALIDATE}(M^{(t)}, \mathcal{D}, m)$
5:    $\{\mathcal{F}_k^{(t)}\}_{k=1}^{K_t} \leftarrow \text{CLUSTER\_FAILURES}(\mathcal{V}^{(t)})$
6:    **for** each failure mode $\mathcal{F}_k^{(t)}$ **do**
7:      construct state $z_{t,k}$ and sample $a_{t,k} \sim \pi_\theta(\cdot \mid z_{t,k})$
8:    **end for**
9:    $\mathcal{D}_{\text{sel}}^{(t)} \leftarrow \text{BUDGETED\_SAMPLE}(\{\mathcal{F}_k^{(t)} : a_{t,k} = 1\}, B_{\text{adv}})$
10:   $\mathcal{D}_{\text{mix}}^{(t)} \leftarrow \text{MIX}(\mathcal{D}, \mathcal{D}_{\text{sel}}^{(t)})$
11:   $M^{(t+1)} \leftarrow \text{Train}(M^{(t)}, \mathcal{D}_{\text{mix}}^{(t)})$
12:   compute reward $G_t$ on $\mathcal{D}_{\text{val}}^{\text{rob}}$ and $\mathcal{D}_{\text{val}}^{\text{clean}}$
13:   update $\pi_\theta$, $R_\phi$, and $b_t$ using $G_t$
14: **end for**
15: **return** $M^{(T)}$

---

**Critic Architecture and Reward Supervision.** The reward model $R_\phi$ is implemented as a lightweight MLP critic operating on failure-mode states rather than individual examples. Its input is the cluster-level state vector $z_{t,k}$ defined above, which contains the cluster size, mean target-model loss, entropy, classification margin, label distribution, retrieval score, judge agreement, novelty score, and previous reward statistics. The critic outputs a scalar utility estimate:

$$R_\phi(z_{t,k}) \in \mathbb{R}, \tag{23}$$

which approximates the expected validation reward obtained by selecting failure mode $\mathcal{F}_k^{(t)}$.

Importantly, we do not compute a separate utility value $\Delta(h)$ for each individual candidate, since doing so would require prohibitively expensive per-example retraining. Instead, reward supervision is defined at the failure-mode level. After the policy selects a subset of failure modes, the target model is retrained once on the resulting adversarial mixture, and the scalar validation reward $G_t$ is computed from robustness improvement, forgetting penalty, and data cost. This same observed return is assigned to all selected failure modes in that round and used to train the critic:

$$\mathcal{L}_R(\phi) = \sum_{k:a_{t,k}=1} \left(R_\phi(z_{t,k}) - G_t\right)^2. \tag{24}$$

The critic is updated once after each retraining round, together with the policy update. Since selection is performed by the stochastic policy $\pi_\theta(a_{t,k} \mid z_{t,k})$, the method does not require a fixed reward threshold $\tau$. This removes the threshold-tuning step used in reward-filtering pipelines and replaces it with validation-driven policy optimization.

## 3.1 Hyperparameter Tuning for Retrieval

Standard training on $P$ may reinforce spurious correlations that fail under distribution shift. Our framework instead concentrates updates on validated failure modes, where such shortcuts are more likely to break and task-relevant reasoning is required. Lemma 3.1 formalizes this intuition by showing that, under stated assumptions, failure-focused sampling reduces shortcut-aligned gradient contributions while increasing semantically meaningful ones. To initialize the hybrid retrieval score, we tune the semantic weight $\alpha$ on 1,000 SNLI training examples using BGE M3. For calibration only, we retrieve a candidate pool of

9 examples per label and treat each pair $(p, x)$ as relevant if $\text{label}(x) = \text{label}(p)$. The hybrid score is: $s_{\text{comb}}(p, x) = \alpha \tilde{s}_{\text{sem}}(p, x) + (1 - \alpha)\tilde{s}_{\text{lex}}(p, x)$. We search $\alpha \in \{0, 0.01, \ldots, 1.0\}$ and select the value with the highest ROC AUC over positive and negative pairs. The best value is $\alpha^* = 0.83$, achieving an AUC of 0.93 in Figure 4. We fix $\alpha = 0.83$ for all downstream experiments. In the main six-shot setting, we use $k = 2$ examples per label. This tuning only initializes retrieval; later adaptation is performed by the contextual-bandit failure-mode policy.

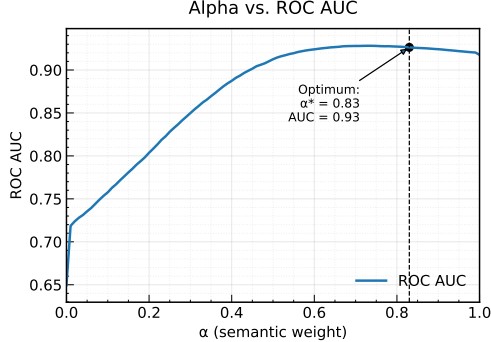

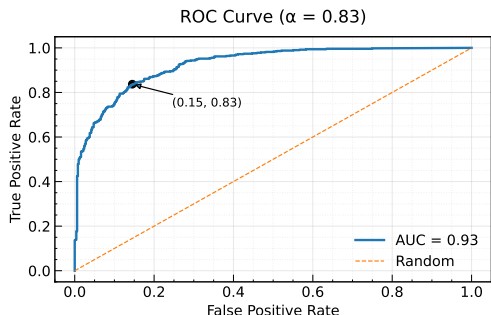

Figure 3: ROC AUC as a function of the semantic-lexical weighting parameter $\alpha$.

Figure 4: ROC curve at optimal $\alpha = 0.83$.

## 3.2 Avoiding Forgetting

Training only on adversarial examples can induce a non-stationary training distribution and lead to catastrophic forgetting, where performance on the original data distribution deteriorates. In our setting, this risk is especially important because the policy is explicitly encouraged to focus on difficult failure modes. To stabilize training, we mix original and selected adversarial examples during retraining. Let $\mathcal{D}_{\text{orig}}$ denote the original training set and $\mathcal{D}_{\text{sel}}^{(t)}$ denote the adversarial examples selected by the policy at iteration $t$. We define the original-to-adversarial mixing ratio:

$$\lambda_{\text{mix}} = \frac{|\mathcal{D}_{\text{orig}}|}{|\mathcal{D}_{\text{sel}}^{(t)}|} \in \left\{0, 1, \frac{1}{2}, \frac{1}{3}, \frac{1}{4}\right\}. \tag{25}$$

Here, $\lambda_{\text{mix}} = 0$ corresponds to training only on selected adversarial examples, while $\lambda_{\text{mix}} = \frac{1}{4}$ denotes one original example per four adversarial examples. For $\lambda_{\text{mix}} > 0$, we construct:

$$\mathcal{D}_{\text{mix}}^{(t)}(\lambda_{\text{mix}}) = \mathcal{D}_{\text{orig}} \cup \text{Sample}\left(\mathcal{D}_{\text{sel}}^{(t)}, \lfloor \lambda_{\text{mix}}^{-1}|\mathcal{D}_{\text{orig}}|\rfloor\right). \tag{26}$$

For $\lambda_{\text{mix}} = 0$, we set: $\mathcal{D}_{\text{mix}}^{(t)}(0) = \mathcal{D}_{\text{sel}}^{(t)}$. For each retrieval mode $m \in \{\text{sem}, \text{lex}, \text{comb}\}$, the target model is optimized for $T$ iterations and evaluated as:

$$A_m(\lambda_{\text{mix}}) = \text{Perf}\left(M^{(T)} \mid \mathcal{D}_{\text{mix}}^{(t)}(\lambda_{\text{mix}})\right), \tag{27}$$

where $\text{Perf}(\cdot)$ denotes the task-specific evaluation metric.

Figure 5 reports $A_{\text{sem}}(\lambda_{\text{mix}})$, $A_{\text{lex}}(\lambda_{\text{mix}})$, and $A_{\text{comb}}(\lambda_{\text{mix}})$ as functions of the mixing ratio. All three curves improve substantially when moderate original-data mixing is introduced. The hybrid retrieval strategy achieves the strongest performance near $\lambda_{\text{mix}} = \frac{1}{4}$, indicating that semantic and lexical retrieval are most effective when policy-selected adversarial failures are balanced with sufficient original-distribution coverage. This setting provides the best trade-off between robustness improvement and forgetting prevention.

We therefore use $\lambda_{\text{mix}}^* = \frac{1}{4}$ in the main experiments. This controlled mixing mitigates catastrophic forgetting while preserving the benefit of failure-aware adversarial training.

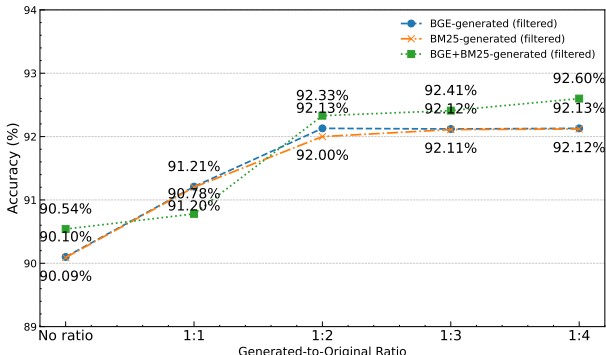

Figure 5: Task performance $A_m(\lambda_{\mathrm{mix}})$ versus the mixing ratio of selected adversarial examples to original training data.

**Theoretical interpretation.** Our method changes the effective training distribution by mixing the original data distribution $P$ with a policy-induced distribution over validated failure modes $\hat{Q}_t^\pi$:

$$P_t^\lambda = (1 - \lambda)P + \lambda\hat{Q}_t^\pi. \tag{28}$$

This view explains why failure-mode curation can reduce reliance on spurious shortcuts: selected failures are examples where the current model's decision rule breaks, so training on them increases the relative contribution of task-relevant gradients. Under the assumptions stated in Appendix B, Lemma B.1 shows that failure-mode sampling reduces shortcut-aligned gradient contributions while preserving core-feature contributions. Propositions B.2 and B.2 further show that the mixture update induces bounded distributional drift and that bounded reward noise causes bounded distortion in the induced sampling policy.

# 4 Evaluation and Results

We evaluate the proposed failure-mode contextual bandit curation pipeline on standard benchmarks for natural language inference and fact verification. All experiments use automatically generated and automatically validated adversarial examples, without additional human annotation.

**Target NLI Model.** For NLI experiments, the target model is `RoBERTa-base-SNLI` (125M parameters) (HuggingFace, 2022), a RoBERTa-base model fine-tuned on SNLI. This model serves as the classifier whose failures are mined, clustered into failure modes, and used for adaptive adversarial retraining.

**Generation LLM.** Adversarial hypotheses are generated using `LLaMA-4-Scout-17B-16E-Instruct` (Meta AI, 2025). For each input, the generator is conditioned on a label-balanced retrieved context constructed using semantic retrieval, lexical retrieval, or hybrid BGE+BM25 retrieval.

**Validation LLMs.** Each generated candidate is automatically validated by an ensemble of three instruction-tuned judge models: `Gemma-3-27B-IT` (Google Research, 2025), `Phi-4` (Microsoft Research, 2025), and `Qwen3-32B` (Qwen Team, 2025). A candidate is retained only when all judges agree with the intended gold label. This validation stage is used to reduce label noise before failure-mode clustering and policy selection.

**Bandit Policy and Critic.** The data curator is implemented as a contextual-bandit policy $\pi_\theta$ over validated failure modes. Each failure mode is represented by a cluster-level state vector containing statistics such as cluster size, target-model loss, entropy, classification margin, label distribution, retrieval score, judge agreement, novelty, and previous reward. The critic $R_\phi$ is a lightweight MLP that predicts the expected validation reward of each selected failure mode. The policy and critic are updated after each retraining round using validation-based feedback.

**Datasets.** We report NLI results on **SNLI** (Bowman et al., 2015), the original human-annotated inference dataset; **ANLI** (Nie et al., 2019), which contains adversarially constructed examples collected through human-and-model interaction; and **MultiNLI** (Williams et al., 2018), a multi-genre corpus for evaluating cross-domain. To assess transfer beyond NLI, we also evaluate on the **FEVER** fact verification benchmark.

To contextualize the gains, we compare against GNLI (Hosseini et al., 2024), a synthetic NLI corpus of approximately 685K LLM-generated examples. Fine-tuning RoBERTa-base on GNLI alone reaches 89.42% on SNLI, 77.07% on ANLI, and 57.61% on MultiNLI. Our pipeline generates approximately 30K adversarial candidates per retrieval strategy, applies target-model failure filtering and automated LLM validation, and retains 6637 BGE-based and 5991 BM25-based candidates for failure-mode clustering and policy-guided sampling. With controlled adversarial mixing, our method improves RoBERTa-base from 88.48% to 92.60% on SNLI, from 75.04% to 80.95% on ANLI, and from 54.67% to 71.99% on MultiNLI, as shown in Table 1. The table also shows that unfiltered adversarial data improves performance, but automated validation and failure-aware selection provide additional gains, indicating that robustness benefits come from prioritizing useful failure modes rather than simply adding more synthetic data.

Table 1: Accuracy (%) on each test set under adversarial mixing. Method names list only *BGE*, *BM25*, and *BGE+BM25*, denoting their respective generation methods. "Reward-Guided" indicates unanimous LLM validation.

| Dataset | RoBERTa Base | Additional Data | Paraphrasing | GNLI | Method | Reward Guided | $r = 0$ | $r = 1$ | $r = 2$ | $r = 3$ | $r = 4$ |
|---|---|---|---|---|---|---|---|---|---|---|---|
| SNLI | 88.48% | 89.42% | 84.73% | - | BGE | No | 90.98% | 91.17% | 91.51% | 91.54% | 91.55% |
| | | | | - | BGE | Yes | 90.10% | 91.21% | 92.13% | 92.12% | 92.13% |
| | | | | - | BM25 | No | 90.03% | 91.02% | 91.14% | 91.18% | 91.19% |
| | | | | - | BM25 | Yes | 90.09% | 91.20% | 92.00% | 92.11% | 92.12% |
| | | | | - | BGE+BM25 | No | 90.11% | 91.19% | 91.35% | 91.61% | 91.68% |
| | | | | - | BGE+BM25 | Yes | 90.54% | 90.78% | 92.33% | 92.41% | **92.60%** |
| | | | | - | T5-Small | - | - | - | - | - | - |
| | | | | - | T5-Large | - | - | - | - | - | - |
| | | | | - | T5-XXL | - | - | - | - | - | - |
| Adversarial NLI | 75.04% | 77.07% | 72.39% | - | BGE | No | 79.07% | 79.72% | 79.52% | 79.92% | 79.47% |
| | | | | - | BGE | Yes | 78.72% | 79.12% | 80.02% | 79.72% | 80.27% |
| | | | | - | BM25 | No | 78.07% | 78.52% | 78.72% | 78.82% | 78.88% |
| | | | | - | BM25 | Yes | 77.97% | 78.62% | 78.92% | 79.07% | 79.12% |
| | | | | - | BGE+BM25 | No | 78.11% | 79.18% | 78.51% | 78.99% | 78.91% |
| | | | | - | BGE+BM25 | Yes | 79.12% | 80.43% | 80.67% | 80.89% | **80.95%** |
| | | | | 33.00% | T5-Small | - | - | - | - | - | - |
| | | | | 45.72% | T5-Large | - | - | - | - | - | - |
| | | | | 57.87% | T5-XXL | - | - | - | - | - | - |
| MultiNLI | 54.67% | 57.61% | 50.01% | - | BGE | No | 69.54% | 69.32% | 70.22% | 69.72% | 71.08% |
| | | | | - | BGE | Yes | 69.15% | 69.62% | 70.22% | 69.97% | 71.15% |
| | | | | - | BM25 | No | 68.34% | 68.72% | 68.92% | 69.22% | 69.54% |
| | | | | - | BM25 | Yes | 68.57% | 68.82% | 69.12% | 69.47% | 69.74% |
| | | | | - | BGE+BM25 | No | 68.05% | 68.15% | 68.81% | 69.11% | 70.02% |
| | | | | - | BGE+BM25 | Yes | 69.21% | 69.37% | 70.59% | 69.81% | **71.99%** |
| | | | | 82.18% | T5-Small | - | - | - | - | - | - |
| | | | | 90.61% | T5-Large | - | - | - | - | - | - |
| | | | | **91.77%** | T5-XXL | - | - | - | - | - | - |

We further evaluate transfer beyond NLI on FEVER. As shown in Table 2, our method improves across model scales. RoBERTa-base reaches 76.58% FEVER score and 79.42% label accuracy, while RoBERTa-large achieves 79.86% FEVER score and 82.45% accuracy. Lightweight models such as SmolLM2-360M and Qwen3-0.6B also benefit, suggesting that failure-aware curation transfers beyond NLI.

Table 2: Comparison on the FEVER benchmark. We report FEVER score and label accuracy.

| Method | Backbone | Dev FEVER | Dev Acc. | Test FEVER | Test Acc. |
|---|---|---|---|---|---|
| GEAR Zhou et al. (2019) | BERT-base | 70.69 | 74.84 | 67.19 | 71.60 |
| KGAT Liu et al. (2020) | RoBERTa-large | 76.11 | 78.29 | 70.38 | 74.07 |
| WgtSum Tymoshenko & Moschitti (2021) | RoBERTa-large | 79.02 | 81.30 | 73.44 | 77.18 |
| BEVERS DeHaven & Scott (2023) | DeBERTa-v2-XL | – | – | 77.70 | 80.24 |
| **Ours** | RoBERTa-base | 81.24 | 83.17 | 76.58 | 79.42 |
| **Ours** | DeBERTa-v3 | 82.91 | 84.76 | 78.12 | 81.03 |
| **Ours** | RoBERTa-large | **84.37** | **86.12** | **79.86** | **82.45** |
| **Ours** | SmolLM2-360M | 79.68 | 81.54 | 74.92 | 77.31 |
| **Ours** | Qwen3-0.6B | 80.97 | 82.83 | 75.89 | 78.64 |

Table 3 and Figure 6 show that increasing the retrieved few-shot context improves performance across SNLI, ANLI, and MultiNLI. In particular, validated BGE reaches 92.15% on SNLI, 80.26% on ANLI, and 71.15%

on MultiNLI in the 9-shot setting, while BM25 shows similar but slightly weaker trends. The 6-shot setting already matches or exceeds the strongest adversarial mixing results, highlighting the importance of retrieval quality and validated failure selection.

Table 3: Few-shot accuracy (%) of our generation methods on each set. Columns indicate the number of few-shot examples.

| Dataset | Method | Filtered? | 0-shot | 3-shot | 6-shot | 9-shot |
|---|---|---|---|---|---|---|
| SNLI | BGE | No | 87.51% | 90.05% | 91.55% | **91.56%** |
| | BGE | Yes | 88.18% | 90.69% | 92.13% | **92.15%** |
| | BM25 | No | 87.51% | 89.69% | 91.19% | 91.22% |
| | BM25 | Yes | 88.18% | 90.67% | **92.12%** | 92.11% |
| | BGE+BM25 | No | 87.51% | 89.98% | **91.68%** | 91.51% |
| | BGE+BM25 | Yes | 88.18% | 90.71% | **92.60%** | 92.51% |
| Adversarial NLI | BGE | No | 75.81% | 77.72% | 79.47% | **79.47%** |
| | BGE | Yes | 76.27% | 78.76% | **80.27%** | 80.26% |
| | BM25 | No | 75.81% | 77.37% | 78.87% | **78.88%** |
| | BM25 | Yes | 76.27% | 77.60% | **79.12%** | 79.10% |
| | BGE+BM25 | No | 75.81% | 77.71% | 78.81% | **78.91%** |
| | BGE+BM25 | Yes | 76.27% | 77.81% | 78.95% | **80.95%** |
| MultiNLI | BGE | No | 67.18% | 69.25% | 71.07% | **71.08%** |
| | BGE | Yes | 67.87% | 69.02% | **71.12%** | 71.15% |
| | BM25 | No | 67.18% | 68.07% | **69.57%** | 69.54% |
| | BM25 | Yes | 67.87% | 68.22% | 69.72% | **69.74%** |
| | BGE+BM25 | No | 67.18% | 69.42% | 69.99% | **70.02%** |
| | BGE+BM25 | Yes | 67.87% | 71.00% | 71.12% | **71.99%** |

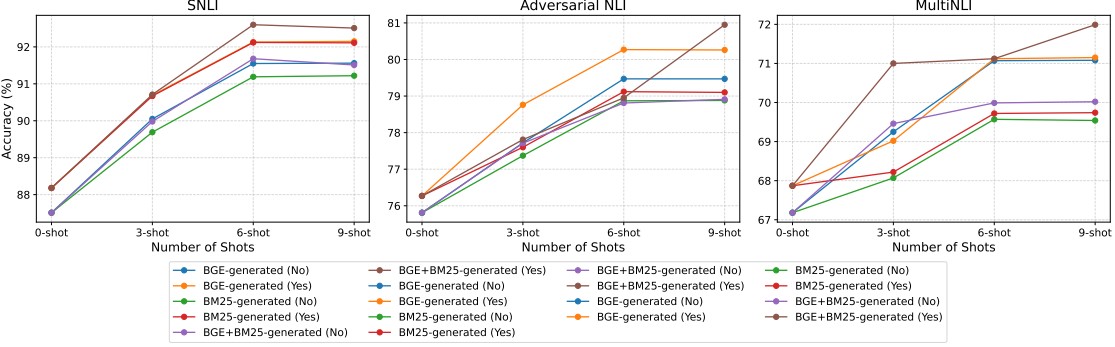

Figure 6: Few-shot accuracy of generation methods by dataset.

## 5 Conclusion and Future Work

We presented a failure-mode contextual bandit framework for adversarial data curation. Instead of selecting synthetic examples with a fixed reward threshold, our method clusters validated model errors into recurring failure modes and learns which modes should be sampled for retraining. This turns the data curator into an adaptive policy that receives validation-based feedback and balances robustness gains, forgetting, and data cost. Across NLI benchmarks and FEVER, the framework improves robustness while using substantially less data than large untargeted synthetic corpora. The results show that prioritizing validated, model-specific failure modes is more effective than simply adding more generated examples. Future work will explore richer failure-mode representations, uncertainty-aware policy updates, adaptive generation budgets, and online curation settings where failures are generated and selected continuously. We also plan to extend the framework to multilingual, domain-specific, and broader robustness tasks, and to combine it with complementary methods such as contrastive learning, adversarial regularization, and representation-level alignment.

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

# A    Appendix

# B    Theoretical Interpretation and Proofs

## B.1    Bias Reduction via Failure-Mode Curation

Let $\mathcal{X}$ be the input space, $Y \in \mathcal{Y}$ the label space, and $P$ the underlying data distribution. Let $f_{\theta_t} : \mathcal{X} \to \Delta(\mathcal{Y})$ denote the target model at iteration $t$, and let $\ell(\theta; x, y)$ be the per-example loss. We define the failure indicator as:

$$e_t(x, y) = \mathbb{I}\left[\arg\max_{c \in \mathcal{Y}} f_{\theta_t}(x)_c \neq y\right]. \tag{29}$$

The corresponding failure probability is:

$$\varepsilon_t = \mathbb{E}_{(x,y) \sim P}\left[e_t(x, y)\right]. \tag{30}$$

When $\varepsilon_t > 0$, the failure-conditioned distribution is:

$$F_t(x, y) = P(x, y \mid e_t(x, y) = 1) = \frac{P(x, y)e_t(x, y)}{\varepsilon_t}. \tag{31}$$

In practice, the method does not sample directly from $F_t$. Instead, it generates candidate adversarial examples, filters candidates that fool $M^{(t)}$, validates them with automated judges, clusters the validated failures into failure modes, and samples from these modes using the contextual-bandit policy $\pi_\theta$. Let $\widehat{F}_t^\pi$ denote the policy-induced distribution over selected validated failure examples. The effective training distribution is:

$$P_t^\lambda = (1 - \lambda)P + \lambda \widehat{F}_t^\pi, \qquad \lambda \in (0, 1). \tag{32}$$

The target model is then updated by stochastic gradient descent on:

$$\mathbb{E}_{(x,y) \sim P_t^\lambda}\left[\ell(\theta; x, y)\right]. \tag{33}$$

**Spurious bias model.** We use an abstract decomposition $x = (c, s)$, where $c$ denotes task-relevant core information and $s$ denotes a spurious feature that is correlated with the label under $P$ but unreliable under distribution shift. This decomposition is not assumed to be explicitly available to the model; it is used only for analysis. Let $g_c(x, y)$ and $g_s(x, y)$ denote the gradient components along unit directions $u_c$ and $u_s$:

$$g_c(x, y) = \langle \nabla_\theta \ell(\theta; x, y), u_c \rangle, \qquad g_s(x, y) = \langle \nabla_\theta \ell(\theta; x, y), u_s \rangle. \tag{34}$$

We assume that standard training on $P$ may reinforce the spurious direction, while failure regions reduce reliance on this shortcut. Specifically, assume there exist constants $\mu_s > 0$ and $\mu_c \geq 0$ such that:

$$\mathbb{E}_P\left[g_s(x, y)\right] \geq \mu_s, \tag{35}$$

and:

$$\mathbb{E}_{F_t}\left[g_s(x, y)\right] \leq 0, \qquad \mathbb{E}_{F_t}\left[g_c(x, y)\right] \geq \mu_c. \tag{36}$$

This captures the intended failure-focused behavior: validated failures are examples where shortcut-based prediction is less reliable and core task-relevant evidence is more important.

**Policy approximation.** The contextual-bandit policy does not need to recover $F_t$ exactly. It is sufficient that the policy-induced selected distribution $\widehat{F}_t^\pi$ approximates $F_t$ with bounded error along the relevant gradient directions:

$$\left| \mathbb{E}_{\widehat{F}_t^\pi} [g_s] - \mathbb{E}_{F_t} [g_s] \right| \leq \delta_s, \qquad \left| \mathbb{E}_{\widehat{F}_t^\pi} [g_c] - \mathbb{E}_{F_t} [g_c] \right| \leq \delta_c, \tag{37}$$

for $\delta_s, \delta_c \geq 0$.

**Lemma A.1. Failure-mode sampling reduces shortcut-aligned gradients.** Under the above assumptions, the mixture distribution $P_t^\lambda$ satisfies:

$$\mathbb{E}_{P_t^\lambda} [g_s(x,y)] \leq (1-\lambda)\mathbb{E}_P [g_s] + \lambda\delta_s, \tag{38}$$

and:

$$\mathbb{E}_{P_t^\lambda} [g_c(x,y)] \geq (1-\lambda)\mathbb{E}_P [g_c] + \lambda(\mu_c - \delta_c). \tag{39}$$

Consequently, if $\delta_s < \mathbb{E}_P[g_s]$, then failure-mode sampling reduces the shortcut-aligned gradient contribution relative to training only on $P$ by at least:

$$\lambda \left( \mathbb{E}_P[g_s] - \delta_s \right). \tag{40}$$

**Proof.** By linearity of expectation under the mixture distribution:

$$\mathbb{E}_{P_t^\lambda} [g_s] = (1-\lambda)\mathbb{E}_P [g_s] + \lambda\mathbb{E}_{\widehat{F}_t^\pi} [g_s]. \tag{41}$$

Using the policy approximation bound and the failure-region assumption:

$$\mathbb{E}_{\widehat{F}_t^\pi} [g_s] \leq \mathbb{E}_{F_t} [g_s] + \delta_s \leq \delta_s. \tag{42}$$

Substituting this into the mixture expression gives:

$$\mathbb{E}_{P_t^\lambda} [g_s] \leq (1-\lambda)\mathbb{E}_P [g_s] + \lambda\delta_s. \tag{43}$$

The result for the core component follows similarly. From the approximation assumption:

$$\mathbb{E}_{\widehat{F}_t^\pi} [g_c] \geq \mathbb{E}_{F_t} [g_c] - \delta_c \geq \mu_c - \delta_c. \tag{44}$$

Therefore:

$$\mathbb{E}_{P_t^\lambda} [g_c] \geq (1-\lambda)\mathbb{E}_P [g_c] + \lambda(\mu_c - \delta_c). \tag{45}$$

Finally, comparing the shortcut bound to $\mathbb{E}_P[g_s]$ gives:

$$\mathbb{E}_P[g_s] - \mathbb{E}_{P_t^\lambda}[g_s] \geq \lambda \left( \mathbb{E}_P[g_s] - \delta_s \right), \tag{46}$$

which is positive whenever $\delta_s < \mathbb{E}_P[g_s]$. This completes the proof.

### B.2 Boundedness of Failure-Aware Bandit Updates

The contextual-bandit policy changes the effective training distribution by selecting different failure modes across iterations. To avoid uncontrolled distributional drift, the target model is trained on a mixture of original and selected adversarial examples. Let $\mathcal{P}_t$ denote the effective training distribution at iteration $t$, and let $\widehat{F}_t^\pi$ denote the distribution induced by the failure-mode policy. We analyze the abstract update:

$$\mathcal{P}_{t+1} = (1 - \eta)\mathcal{P}_t + \eta\widehat{F}_t^\pi, \qquad \eta \in (0, 1). \tag{47}$$

**Proposition A.2. Per-step distributional drift is bounded.** For any iteration $t$, the per-step change in the effective training distribution satisfies:

$$\|\mathcal{P}_{t+1} - \mathcal{P}_t\|_1 = \eta \left\|\widehat{F}_t^\pi - \mathcal{P}_t\right\|_1 \leq 2\eta. \tag{48}$$

Moreover, for any $t$, the distance from the initial distribution is bounded by:

$$\|\mathcal{P}_t - \mathcal{P}_0\|_1 \leq 2. \tag{49}$$

**Proof.** From the update rule:

$$\mathcal{P}_{t+1} - \mathcal{P}_t = \eta \left(\widehat{F}_t^\pi - \mathcal{P}_t\right). \tag{50}$$

Taking the $\ell_1$ norm gives:

$$\|\mathcal{P}_{t+1} - \mathcal{P}_t\|_1 = \eta \left\|\widehat{F}_t^\pi - \mathcal{P}_t\right\|_1. \tag{51}$$

Because the $\ell_1$ distance between any two probability distributions is at most 2:

$$\|\mathcal{P}_{t+1} - \mathcal{P}_t\|_1 \leq 2\eta. \tag{52}$$

The second statement follows from the same fact, since both $\mathcal{P}_t$ and $\mathcal{P}_0$ are probability distributions:

$$\|\mathcal{P}_t - \mathcal{P}_0\|_1 \leq 2. \tag{53}$$

This establishes that each update is locally controlled by the mixing coefficient $\eta$.

**Reward-noise setting.** The policy is updated using validation feedback, which may be noisy due to finite validation sets and stochastic retraining. Let $u_k$ denote the ideal utility of failure mode $k$, and let the observed utility be:

$$\tilde{u}_k = u_k + \xi_k, \qquad |\xi_k| \leq \varepsilon. \tag{54}$$

For analysis, consider the normalized policy-induced allocation distribution over failure modes:

$$\pi_u(k) = \frac{\exp(u_k)}{\sum_j \exp(u_j)}, \qquad \pi_{\tilde{u}}(k) = \frac{\exp(\tilde{u}_k)}{\sum_j \exp(\tilde{u}_j)}. \tag{55}$$

This log-linear allocation is a standard smooth relaxation of selecting failure modes according to estimated utility.

**Proposition A.3. Bounded reward noise induces bounded sampling distortion.** Assume $|\xi_k| \leq \varepsilon$ for all failure modes $k$. Then, for any set of failure modes $A$ with $\pi_u(A) > 0$:

$$e^{-2\varepsilon} \leq \frac{\pi_{\tilde{u}}(A)}{\pi_u(A)} \leq e^{2\varepsilon}. \tag{56}$$

**Proof.** For each failure mode $k$:

$$e^{-\varepsilon} \exp(u_k) \leq \exp(\tilde{u}_k) \leq e^{\varepsilon} \exp(u_k). \tag{57}$$

Summing over all modes gives:

$$e^{-\varepsilon} \sum_j \exp(u_j) \leq \sum_j \exp(\tilde{u}_j) \leq e^{\varepsilon} \sum_j \exp(u_j). \tag{58}$$

Combining the numerator and denominator bounds yields, for each $k$:

$$e^{-2\varepsilon} \leq \frac{\pi_{\tilde{u}}(k)}{\pi_u(k)} \leq e^{2\varepsilon}. \tag{59}$$

Summing over all $k \in A$ preserves the same multiplicative bound:

$$e^{-2\varepsilon} \leq \frac{\sum_{k \in A} \pi_{\tilde{u}}(k)}{\sum_{k \in A} \pi_u(k)} \leq e^{2\varepsilon}. \tag{60}$$

Therefore:

$$e^{-2\varepsilon} \leq \frac{\pi_{\tilde{u}}(A)}{\pi_u(A)} \leq e^{2\varepsilon}. \tag{61}$$

Together, Proposition B.2 and Proposition B.2 show that the failure-aware bandit update operates in a controlled regime: the mixture coefficient bounds the per-step distributional shift, and bounded noise in validation-based utility estimates induces only bounded distortion in the policy-induced failure-mode allocation.

## B.3 Backbone and Training Strategy Ablation

We analyze the impact of backbone architecture and training strategy using the results reported in Table 4. This experiment evaluates five representative models spanning a wide range of parameter scales, from lightweight decoder-only architectures (SmolLM2-360M and Qwen3-0.6B) to large encoder-based models (RoBERTa-base, DeBERTa-v3, and RoBERTa-large). For each backbone, we compare three settings: (i) evaluation without fine-tuning (No FT), (ii) fine-tuning with paraphrase-based data augmentation (Paraphrasing), and (iii) our reinforcement-guided failure-driven training framework (Ours). All models are trained under identical conditions using fixed 6-shot prompting and a constant original-to-generated ratio of 1:4 ($r = 4$).

Several consistent trends emerge across all datasets. First, models evaluated without fine-tuning exhibit the lowest performance in every configuration, confirming that direct transfer without adaptation is insufficient for robust NLI. Paraphrase-based augmentation yields moderate improvements over No FT, indicating that generic linguistic variation helps alleviate some distributional mismatch. However, these gains remain limited, particularly on challenging benchmarks such as Adversarial NLI, where paraphrasing fails to systematically target the model's dominant failure modes.

In contrast, our reinforcement-guided approach consistently achieves the strongest performance for all backbone architectures and datasets. On SNLI, our method improves RoBERTa-base from 90.72% under paraphrasing to 92.60%, and yields comparable gains for smaller models such as SmolLM2-360M (+1.81 points) and Qwen3-0.6B (+1.91 points). Similar patterns are observed on Adversarial NLI, where our framework outperforms paraphrasing by 2.83 points for RoBERTa-base and by more than 2 points for all other models. On MultiNLI, which exhibits substantial genre diversity, reinforcement-guided training produces consistent improvements ranging from 2.36 to 3.47 points over paraphrasing.

The results further demonstrate that the benefits of failure-driven policy learning are preserved across model scales. While larger backbones such as RoBERTa-large achieve higher absolute accuracy, smaller and medium-sized models also benefit substantially from targeted adversarial mining. This indicates that the proposed framework does not rely on excess model capacity, but instead improves generalization by reshaping the training distribution toward informative failure regions.

Importantly, the consistent gap between paraphrasing and our method highlights the limitations of heuristic data augmentation. Paraphrase-based approaches introduce surface-level variation but do not adaptively concentrate on systematic errors. In contrast, our method leverages reinforcement learning to prioritize samples that expose decision boundary weaknesses, resulting in more efficient and targeted learning.

Overall, the results in Table 4 demonstrate that reinforcement-guided adversarial training yields robust and scalable improvements across diverse architectures and training regimes, confirming the generality of the proposed approach.

**Effect of Generator and Verifier Scale.** Our framework relies on large language models for adversarial generation and validation, and its performance may depend on their representational capacity. To analyze this dependence, future work will systematically vary the scale of both the generator and verifier models, ranging from lightweight open-source LLMs to large proprietary systems. Such controlled experiments will enable a principled assessment of how robustness gains trade off against computational cost, and will clarify the operating regimes in which reinforcement-guided data selection remains effective under limited budgets.

**Disentangling Generation and Verification Contributions.** An important open question concerns the relative contribution of the generation and verification components. While both modules are optimized through policy feedback, their individual roles in driving performance gains are not yet fully disentangled. A promising direction is to decouple these stages by fixing one component while varying the capacity of the other, thereby isolating the effect of generation quality versus validation reliability. This analysis would help determine whether improvements primarily stem from producing more challenging candidates or from more accurate reward estimation via verification.

Table 4: Accuracy (%) across different backbone models and training strategies. "No FT" denotes evaluation without fine-tuning, "Paraphrasing" denotes augmentation via paraphrase-based data, and "Ours" denotes reinforcement-guided training. All experiments use fixed 6-shot prompting and an original-to-generated data ratio of 1:4 ($r = 4$).

| Dataset | SmolLM2-360M | | | Qwen3-0.6B | | | RoBERTa-base | | | DeBERTa-v3 | | | RoBERTa-large | | |
|---|---|---|---|---|---|---|---|---|---|---|---|---|---|---|---|
| | No FT | Para | Ours | No FT | Para | Ours | No FT | Para | Ours | No FT | Para | Ours | No FT | Para | Ours |
| SNLI | 79.42 | 82.36 | 84.17 | 82.95 | 85.41 | 87.32 | 88.48 | 90.72 | 92.60 | 87.91 | 89.84 | 91.94 | 89.37 | 91.12 | 93.21 |
| Adversarial NLI | 67.18 | 70.04 | 72.48 | 70.93 | 73.41 | 75.91 | 75.04 | 78.12 | 80.95 | 74.26 | 77.03 | 79.87 | 75.88 | 78.94 | 81.62 |
| MultiNLI | 61.94 | 64.37 | 66.73 | 64.82 | 67.05 | 69.48 | 54.67 | 68.42 | 71.99 | 66.71 | 69.12 | 71.42 | 68.09 | 70.33 | 72.86 |

## B.4 Ablation Study: Effect of the Contextual-Bandit Policy

We analyze the contribution of the contextual-bandit curator on SNLI by replacing the learned failure-mode policy with several alternatives. All variants use the same backbone, retrieval strategy, automated validation, adversarial budget, and retraining protocol. The full model achieves 92.60% accuracy on SNLI.

Let $\{\mathcal{F}_1^{(t)}, \ldots, \mathcal{F}_{K_t}^{(t)}\}$ denote the validated failure-mode clusters at iteration $t$. Each cluster is represented by a state vector $z_{t,k}$, and the curator selects actions $a_{t,k} \in \{0,1\}$ indicating whether failure mode $\mathcal{F}_k^{(t)}$ is sampled for retraining.

**Full Model: Contextual-Bandit Policy.** The proposed model uses a stochastic policy $\pi_\theta$ over failure modes:

$$\pi_\theta(a_{t,k} = 1 \mid z_{t,k}) = \sigma(f_\theta(z_{t,k})). \tag{62}$$

After retraining, the policy receives validation reward $G_t$, which balances robustness gain, forgetting, and data cost. A critic $R_\phi$ estimates the expected return of each selected failure mode:

$$R_\phi(z_{t,k}) \approx \mathbb{E}[G_t \mid z_{t,k}]. \tag{63}$$

**Random Failure-Mode Policy.** This baseline removes learned selection. Failure modes are sampled uniformly under the same adversarial budget:

$$a_{t,k} \sim \mathrm{Bernoulli}(p). \tag{64}$$

**Heuristic Failure-Mode Policy.** This variant replaces $\pi_\theta$ with a deterministic uncertainty-based rule. Failure modes are ranked by the mean predictive entropy of the target model:

$$s(\mathcal{F}_k^{(t)}) = \frac{1}{|\mathcal{F}_k^{(t)}|} \sum_{(x,o,y) \in \mathcal{F}_k^{(t)}} H\left(M^{(t)}(\cdot \mid x, o)\right). \tag{65}$$

The highest-scoring clusters are selected until the adversarial budget is reached.

**Frozen Policy and Critic.** Here, $\pi_\theta$ and $R_\phi$ are initialized in the first round and then kept fixed. This tests whether continual validation-based adaptation is necessary.

**No Failure-Mode Clustering.** This baseline removes the failure-mode abstraction and samples validated failures directly. It preserves target-model filtering and automated validation but does not group failures into recurring modes.

**Oracle Failure-Mode Policy.** As an upper bound, we approximate the utility of a failure mode using its observed validation improvement after retraining:

$$s_{\mathrm{oracle}}(\mathcal{F}_k^{(t)}) = \mathrm{Perf}(M_k^{(t+1)}) - \mathrm{Perf}(M^{(t)}), \tag{66}$$

where $M_k^{(t+1)}$ denotes a model retrained using samples from failure mode $\mathcal{F}_k^{(t)}$. This variant is not deployable because it requires separate retraining for each candidate failure mode.

Table 5 reports the results. Random and heuristic policies underperform the full model, showing that static selection is insufficient. Freezing the policy and critic also degrades performance, indicating that adaptation across rounds is important. Removing failure-mode clustering further reduces accuracy, confirming that selecting recurring failure types is more effective than selecting isolated examples.

## B.5 Ablation with Heuristic Failure-Mode Policies

To assess whether learned policy optimization is necessary, we replace the contextual-bandit policy $\pi_\theta$ with several heuristic failure-mode selection rules. These baselines use the same generated candidates, target-

Table 5: Ablation of the contextual-bandit curator on SNLI.

| Method | Failure Modes | Adaptive Policy | Critic $R_\phi$ | Accuracy (%) |
|---|---|---|---|---|
| Full (Ours) | ✓ | ✓ | ✓ | 92.60 |
| Random Failure-Mode Policy | ✓ | ✗ | ✗ | 89.70 |
| Heuristic Entropy Policy | ✓ | ✗ | ✗ | 90.45 |
| Frozen Policy and Critic | ✓ | ✗ | ✓ | 91.20 |
| No Failure-Mode Clustering | ✗ | ✓ | ✓ | 90.85 |
| Oracle Failure-Mode Policy | ✓ | GT | GT | 93.40 |

model failure filtering, automated LLM validation, retrieval weight $\alpha = 0.83$, and mixing ratio $\lambda_{\mathrm{mix}} = \frac{1}{4}$ as the full method. The only difference is how validated failure modes are selected for retraining.

Let $\mathcal{F}_k^{(t)}$ denote a validated failure-mode cluster at iteration $t$. Each heuristic assigns a cluster-level score $s(\mathcal{F}_k^{(t)})$, and clusters are selected in descending order until the adversarial budget is reached.

**Confidence-Based Policy.** This policy prioritizes clusters where the target model has low predictive confidence:

$$s_{\mathrm{conf}}(\mathcal{F}_k^{(t)}) = \frac{1}{|\mathcal{F}_k^{(t)}|} \sum_{(x,o,y) \in \mathcal{F}_k^{(t)}} \left( 1 - \max_{c \in \mathcal{Y}} M^{(t)}(c \mid x, o) \right). \tag{67}$$

**Loss-Based Policy.** This policy selects clusters that induce high average supervised loss:

$$s_{\mathrm{loss}}(\mathcal{F}_k^{(t)}) = \frac{1}{|\mathcal{F}_k^{(t)}|} \sum_{(x,o,y) \in \mathcal{F}_k^{(t)}} \ell(M^{(t)}(x, o), y). \tag{68}$$

**Margin-Based Policy.** This policy prioritizes clusters with small separation between the top two predicted classes:

$$s_{\mathrm{margin}}(\mathcal{F}_k^{(t)}) = \frac{1}{|\mathcal{F}_k^{(t)}|} \sum_{(x,o,y) \in \mathcal{F}_k^{(t)}} \left( 1 - \left[ p_\theta^{(1)}(x, o) - p_\theta^{(2)}(x, o) \right] \right), \tag{69}$$

where $p_\theta^{(1)}(x, o)$ and $p_\theta^{(2)}(x, o)$ are the highest and second-highest predicted class probabilities.

**Learned Bandit Policy.** The full method uses the learned contextual-bandit policy:

$$\pi_\theta(a_{t,k} = 1 \mid z_{t,k}) = \sigma(f_\theta(z_{t,k})), \tag{70}$$

where $z_{t,k}$ includes loss, entropy, margin, label distribution, retrieval score, judge agreement, novelty, cluster size, and previous reward statistics. Unlike the heuristic policies, $\pi_\theta$ is updated using validation reward $G_t$ after retraining.

Table 6: Comparison of heuristic failure-mode policies and the learned contextual-bandit policy.

| Selection Policy | SNLI | ANLI | MultiNLI |
|---|---|---|---|
| Confidence-Based Policy | 90.84 | 78.12 | 69.21 |
| Loss-Based Policy | 91.02 | 78.45 | 69.68 |
| Margin-Based Policy | 90.67 | 77.94 | 68.97 |
| Learned Bandit Policy $\pi_\theta$ | **92.60** | **80.95** | **71.99** |

Heuristic policies capture only instantaneous model uncertainty or training difficulty. As a result, they may oversample noisy, redundant, or locally difficult failures that do not produce sustained validation gains. In contrast, the learned contextual-bandit policy is optimized using downstream validation feedback and can adapt across curation rounds. The consistent improvement over heuristic policies shows that adaptive failure-mode selection is more effective than static uncertainty-based selection.

## B.6 Component Analysis of Failure-Mode Curation

Table 7 evaluates the contribution of the main components in the proposed failure-mode contextual bandit curation framework. The full method achieves the best performance across all benchmarks, reaching 92.60% on SNLI, 80.95% on ANLI, and 71.99% on MultiNLI. This confirms that combining retrieval-augmented generation, target-model failure filtering, automated validation, failure-mode clustering, contextual-bandit selection, and controlled original-data mixing provides the strongest robustness gains.

Removing the contextual-bandit policy substantially reduces performance. Random cluster selection performs considerably worse, especially on MultiNLI, indicating that not all failure modes are equally useful for retraining. Selecting clusters by top loss improves over random selection, but remains below the full method, showing that simple difficulty-based heuristics are less effective than validation-driven policy learning. Similarly, replacing failure-mode selection with per-example selection also degrades performance, suggesting that grouping failures into recurring modes provides a more stable and useful unit for adversarial data curation.

The ablations further show that automated judge validation and target-model failure filtering are important for maintaining data quality. Without judge validation, performance drops across all datasets, indicating that noisy or incorrectly labeled generated examples can weaken the retraining signal. Removing failure filtering causes an even larger degradation, showing that explicitly focusing on examples that expose current model errors is central to the proposed approach.

Finally, the retrieval and mixing ablations demonstrate the importance of both informative generation context and forgetting control. Removing retrieved context reduces performance, confirming that retrieved few-shot examples help guide the generator toward more useful adversarial candidates. Training without original-data mixing also hurts performance, supporting the need to balance selected adversarial failures with original training examples in order to improve robustness while limiting forgetting.

Table 7: Ablation study of the proposed failure-mode contextual bandit curation framework.

| Method | SNLI | ANLI | MultiNLI |
|---|---|---|---|
| **Full method** | **92.60** | **80.95** | **71.99** |
| No bandit, random clusters | 89.70 | 76.50 | 66.20 |
| No bandit, top-loss clusters | 91.02 | 78.45 | 69.68 |
| No clustering, per-example selection | 90.85 | 78.20 | 69.10 |
| No judge validation | 91.68 | 78.91 | 70.02 |
| No failure filtering | 89.15 | 76.90 | 66.50 |
| No retrieved context (0-shot) | 88.18 | 76.27 | 67.87 |
| No original-data mixing | 90.54 | 79.12 | 69.21 |

## B.7 Judge Ensemble Configuration

With the retrieval weight fixed at $\alpha = 0.83$ and the generated-to-original example ratio set to 1:4, we evaluated the impact of varying the number of "judges" (independent LLM validators) on downstream accuracy. All experiments were run on the SNLI test set. We filtered examples by requiring unanimous agreement among the selected judges and then measured classification accuracy on the remaining items.

Table 8: Filtering and accuracy under different judge ensemble sizes (SNLI test, 1:4 gen:orig, $\alpha = 0.83$). Judges: G = Gemma-3-27B-IT (Google Research, 2025), Q = Qwen3-32B (Qwen Team, 2025), P = Phi-4 (Microsoft Research, 2025).

| # Judges | # Examples | Accuracy (%) | Judges |
|---|---|---|---|
| 1 | 16,147 | 91.02 | G |
| 2 | 9,312 | 91.49 | G + Q |
| 3 | 6,438 | **92.13** | G + Q + P |

As shown in Table 8 and Figure 7, the three-judge ensemble yields the highest accuracy (92.13%) on 6,438 filtered observations. Both the two-judge and single-judge configurations retain more examples but achieve lower accuracies of 91.49% (9,312 examples) and 91.02% (16,147 examples), respectively. Gemma-3-27B-IT consistently remains in all configurations, with Qwen3-32B joining for the two-judge setup and Phi-4 for the three-judge ensemble. We adopt the three-judge configuration for all subsequent evaluations.

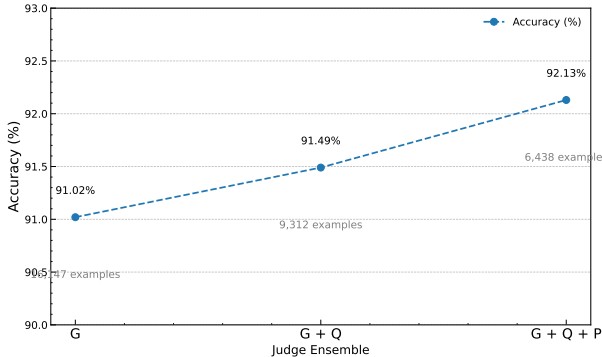

Figure 7: Accuracy vs. number of judges (SNLI test, $\alpha = 0.83$, 1:4 generated:original). Points are annotated with the number of filtered examples.

### B.7.1 Evaluation with Small Judge Models

To study the robustness of our validation pipeline under weaker supervision, we additionally evaluated the judge ensemble using lightweight language models, including Phi-2 Javaheripi et al. (2023), Qwen2.5-1.5B Yang et al. (2024), and SmolLM2-360M Allal et al. (2025). All experiments were conducted using the same retrieval weight ($\alpha = 0.83$) and generated-to-original ratio (1:4) as in Table 6.

We followed the same filtering protocol, retaining only examples for which all selected judges unanimously agreed. Classification accuracy was then measured on the remaining test instances. Table 9 reports the results on the SNLI test set.

Table 9: Filtering and accuracy under different small judge ensemble sizes (SNLI test, 1:4 gen:orig, $\alpha = 0.83$). Judges: S = SmolLM2-360M, P = Phi-2, Q = Qwen2.5-1.5B.

| # Judges | # Examples | Accuracy (%) | Judges |
|---|---|---|---|
| 1 | 17,982 | 89.74 | S |
| 2 | 11,436 | 90.21 | S + P |
| 3 | 7,905 | 90.96 | S + P + Q |

Compared to large-model ensembles (Table 6), small judge models yield moderately lower accuracy and weaker filtering precision. Nevertheless, performance improves consistently with ensemble size, and even a single lightweight judge provides substantial robustness gains. These results indicate that our framework

degrades gracefully under weaker validation models, supporting its applicability in low-resource and cost-constrained settings.

## B.8 Dataset Comparison

To gain insights into the relationship between the data generated in our experiment and existing benchmarks, we first extracted the 10 most frequent non-stopwords from each dataset. This qualitative analysis highlights topical overlap and domain shifts. To quantify similarity more rigorously, we computed two complementary metrics across seven collections-SNLI Train, BGE-generated, BM25-generated, SNLI Test, Adversarial NLI, Multi-NLI, and our hybrid BGE+BM25-generated set: TF-IDF cosine similarity and BERTScore F1 (Zhang et al., 2020).

**TF-IDF Cosine Similarity.** Let each dataset $D$ be represented by a TF-IDF vector $\mathbf{v}_D \in \mathbb{R}^n$, where $n$ is the vocabulary size and the $i$th component is

$$v_{D,i} = \mathrm{TF}_{D,i} \cdot \log\left(\tfrac{N}{\mathrm{DF}_i}\right),$$

with $\mathrm{TF}_{D,i}$ the term frequency in $D$, $N$ the total number of datasets, and $\mathrm{DF}_i$ the number of datasets containing term $i$. We then define

$$\mathrm{sim}_{\mathrm{TFIDF}}(D, D') = \frac{\mathbf{v}_D \cdot \mathbf{v}_{D'}}{\|\mathbf{v}_D\| \, \|\mathbf{v}_{D'}\|}.$$

Figure 8 shows the resulting $7 \times 7$ matrix. Notably, the hybrid BGE+BM25 set has a TF-IDF similarity of approximately 0.0251 with SNLI Train, 0.0188 with SNLI Test, and 0.0150 with Multi-NLI-intermediate between its BGE-only and BM25-only counterparts.

**BERTScore F1.** We next measure semantic overlap by applying BERTScore F1, which aligns token embeddings from a pre-trained transformer and computes an $F_1$ score:

$$\mathrm{P} = \frac{1}{|x|} \sum_{t \in x} \max_{s \in y} \cos(\mathbf{e}_t, \mathbf{e}_s), \mathrm{R} = \frac{1}{|y|} \sum_{s \in y} \max_{t \in x} \cos(\mathbf{e}_s, \mathbf{e}_t),$$

$$\mathrm{F1} = 2 \cdot \frac{\mathrm{P}\,\mathrm{R}}{\mathrm{P} + \mathrm{R}},$$

where $x, y$ are token sequences from two datasets and $\mathbf{e}$ are contextual embeddings. Figure 9 displays the $7 \times 7$ BERTScore F1 matrix. The hybrid set scores about 0.8658 with SNLI Train, 0.8534 with SNLI Test, 0.8458 with Adversarial NLI, and 0.8554 with Multi-NLI, again falling between its BGE-only and BM25-only pairs. These results confirm that our validated adversarial examples share both lexical and semantic patterns with standard NLI benchmarks, while still introducing novel, challenging variations.

From Figure 8, we see that both BGE- and BM25-generated data share moderate lexical overlap with the original SNLI Train set (cosine similarities around 0.02-0.03), but diverge more substantially from the Adversarial NLI and Multi-NLI benchmarks. In contrast, Figure 9 shows that semantically these generated datasets align much more closely with SNLI Train and SNLI Test (BERTScore F1 values above 0.85), indicating that although the surface vocabulary varies, the core contextual meaning is well preserved.

## B.9 Generated Dataset Characteristics and Hypothesis Lengths

We first examined the most frequent tokens in each corpus to identify thematic patterns. In the `SNLI train` (Bowman et al., 2015) and `SNLI test` (Bowman et al., 2015) sets, words like "man," "woman," and "people" dominate, reflecting descriptions of social interactions. The `Adversarial NLI` dataset (Nie et al., 2019) shifts focus to media and chronology, with top tokens such as "film," "first," and "scene," while the `Multi-NLI test` set (Williams et al., 2018) uses more abstract, domain-diverse language-terms like "author," "context," and "claim" appear frequently.

Turning to our three LLM-generated sets-Generated-BM25,

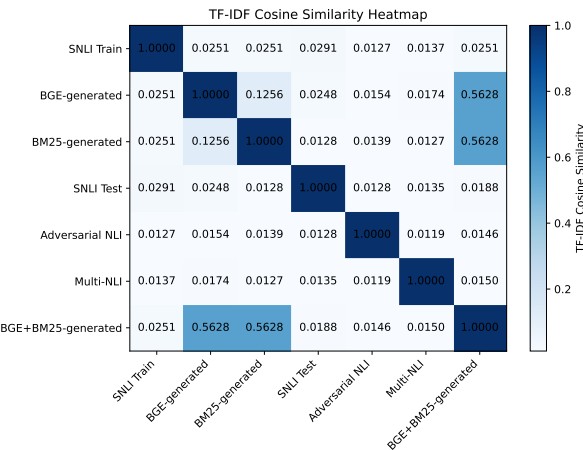

Figure 8: Pairwise TF-IDF cosine similarity between datasets.

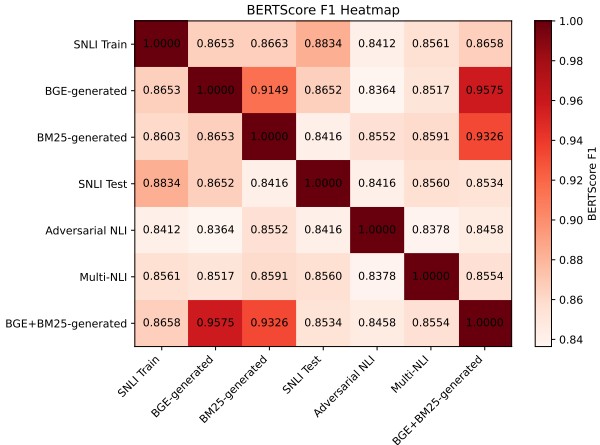

Figure 9: Pairwise BERTScore F1 between datasets.

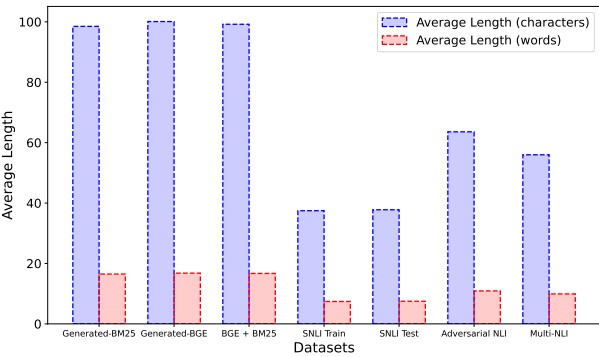

Figure 10: Comparison of average hypothesis lengths (in characters and words) across datasets: `Generated-BM25`, `Generated-BGE`, `SNLI train` (Bowman et al., 2015), `SNLI test` (Bowman et al., 2015), `Adversarial NLI` (Nie et al., 2019), and `Multi-NLI` (Williams et al., 2018).

Generated-BGE and BGE+BM25-we again see a high incidence of speculative and gender-related terms ("could," "would," "woman," "he," "she"), confirming that all retrieval strategies surface similar thematic content with only minor stylistic differences.

Figure 10 compares the average hypothesis lengths across all seven datasets. Each of the generated sets produces the longest hypotheses-around 98-100 characters (16-17 words)-demonstrating the LLM's tendency toward more elaborate constructions when given rich few-shot contexts. By contrast, the `SNLI train` and `SNLI test` annotations remain quite concise ($\approx$ 37-38 characters, 7-8 words), reflecting the brevity of human-written examples. The `Adversarial NLI` instances average $\approx$ 64 characters (11 words), and the `Multi-NLI` examples average $\approx$ 56 characters (10 words), underscoring their intermediate complexity. These length patterns highlight how our adversarial RAG pipeline generates richer, more challenging hypotheses while preserving diversity across data sources.

### B.10 Retrieval Accuracy Across Similarity Metrics

For purely lexical retrieval we employ BM25 with parameters $k_1 = 1.5$ and $b = 0.75$. The BM25 score for a query $p$ and document $x$ is given by

$$s_{\text{BM25}}(p, x) = \sum_{t \in p} \text{IDF}(t) \frac{\text{tf}(t, x)(k_1 + 1)}{\text{tf}(t, x) + k_1 \left(1 - b + b \frac{|x|}{\text{avgdl}}\right)}, \tag{71}$$

and for each label $y'$ we retrieve the top-$k$ documents

$$\mathcal{C}_p^{\text{lex}}(y') = \arg \max_{\substack{S \subseteq \mathcal{D}_{y'} \\ |S| = k}} \sum_{x \in S} s_{\text{BM25}}(p, x). \tag{72}$$

For embedding-based retrieval, we first compute cosine similarity

$$S_{\text{cos}}(E_I, E_{\mathcal{D}}) = \frac{E_I \cdot E_{\mathcal{D}}}{\|E_I\|_2 \|E_{\mathcal{D}}\|_2}, \tag{73}$$

and raw dot product

$$S_{\text{dp}}(E_I, E_{\mathcal{D}}) = E_I \cdot E_{\mathcal{D}} = \sum_{i=1}^{d} (E_I)_i (E_{\mathcal{D}})_i. \tag{74}$$

We additionally assess two norm-based distances: the $L_2$ distance

$$d_2(E_I, E_{\mathcal{D}}) = \|E_I - E_{\mathcal{D}}\|_2 = \sqrt{\sum_{i=1}^{d} \left((E_I)_i - (E_{\mathcal{D}})_i\right)^2}, \tag{75}$$

and the $L_1$ distance

$$d_1(E_I, E_{\mathcal{D}}) = \|E_I - E_{\mathcal{D}}\|_1 = \sum_{i=1}^{d} \left|(E_I)_i - (E_{\mathcal{D}})_i\right|. \tag{76}$$

Finally, to capture distributional discrepancies we examine the Bray-Curtis distance

$$d_{\text{BC}}(E_I, E_{\mathcal{D}}) = \frac{\sum_{i=1}^{d} \left|(E_I)_i - (E_{\mathcal{D}})_i\right|}{\sum_{i=1}^{d} \left|(E_I)_i + (E_{\mathcal{D}})_i\right|}, \tag{77}$$

and the Canberra distance

$$d_{\text{Can}}(E_I, E_{\mathcal{D}}) = \sum_{i=1}^{d} \frac{\left|(E_I)_i - (E_{\mathcal{D}})_i\right|}{\left|(E_I)_i\right| + \left|(E_{\mathcal{D}})_i\right|}. \tag{78}$$

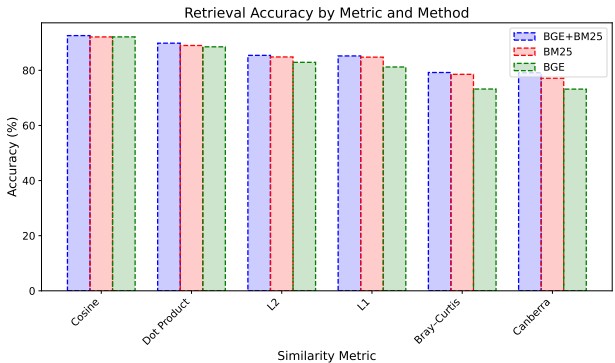

Figure 11: Retrieval accuracy (%) by similarity metric for BGE+BM25, BM25, and BGE.

Figure 11 demonstrates that BGE+BM25 outperforms both BM25 alone and BGE alone across all six metrics, achieving 92.60% (cosine), 89.85% (dot product), 85.43% ($L_2$), 85.22% ($L_1$), 79.21% (Bray-Curtis) and 79.12% (Canberra). Pure BM25 and pure BGE match closely on cosine but degrade more sharply on norm- and distribution-based distances, confirming the robustness of the hybrid lexical-semantic approach.

### B.11 Hyperparameter Optimization and Reproducibility

To ensure fair and reproducible evaluation, all target models are fine-tuned using a standardized hyperparameter optimization protocol. We employ Bayesian optimization via Optuna to search over learning and regularization parameters, using validation accuracy as the objective.

**Tokenization and Input Representation.** All premise–hypothesis pairs are tokenized using the RoBERTa tokenizer with a maximum sequence length of 128. Inputs are padded and truncated to fixed length to ensure consistent batch construction across runs. Each example is represented by input IDs, attention masks, and class labels.

**Training and Evaluation Splits.** For efficiency during hyperparameter tuning, we use the full augmented training set and a fixed validation subset of 1,200 examples. Samples with undefined labels are removed prior to evaluation. All datasets are formatted in PyTorch tensors.

**Search Space.** We optimize the following hyperparameters:

$$\eta \sim \text{LogUniform}(10^{-6}, 10^{-4}), \tag{79}$$
$$E \sim \{1, 2, 3, 5\}, \tag{80}$$
$$B \sim \{1, 2, 4, 8, 16\}, \tag{81}$$
$$\lambda \sim \text{Uniform}(10^{-4}, 10^{-2}), \tag{82}$$

where $\eta$ denotes the learning rate, $E$ the number of training epochs, $B$ the per-device batch size, and $\lambda$ the weight decay coefficient.

**Optimization Procedure.** For each trial, a RoBERTa-based classifier is fine-tuned using the HuggingFace `Trainer` framework. Models are evaluated at the end of each epoch, and the best-performing checkpoint is retained based on validation accuracy. Early stopping is implicitly enforced by selecting the best epoch. We perform 40 independent trials and select the configuration that maximizes validation accuracy.

All experiments are conducted on a single NVIDIA A100 GPU. Each training epoch requires approximately 3.11 minutes on average.

**Evaluation Metric.** All hyperparameter configurations are evaluated using classification accuracy:

$$\text{Acc} = \frac{1}{N} \sum_{i=1}^{N} \mathbb{I}\left[\hat{y}_i = y_i\right], \tag{83}$$

where $\hat{y}_i$ and $y_i$ denote predicted and ground-truth labels for sample $i$, respectively.

**Reproducibility Measures.** To reduce variance across runs, we fix random seeds for data sampling, model initialization, and optimization. All experiments use identical preprocessing, prompt templates, and evaluation splits. Hyperparameter search spaces, optimization budgets, and validation subsets are fully specified to enable exact replication of our results.

The complete training and optimization scripts will be released upon publication.

### B.12 Illustrative Example: Failure-Mode Bandit Curation for NLI

Figure 12 illustrates one iteration of the proposed framework on a Natural Language Inference (NLI) example. The example is intended to show the operational flow of the method and does not represent a full training run.

Given a premise and target label, the generator produces multiple candidate hypotheses conditioned on retrieved few-shot examples. The current target model first filters these candidates by retaining only those that induce an incorrect prediction. The remaining candidates are then checked by an automated LLM judge ensemble to ensure label consistency.

Validated failures are embedded and grouped into failure-mode clusters. A contextual-bandit policy observes the state of each failure mode and selects which modes should be sampled under the adversarial budget. The selected examples are mixed with original training data and used to retrain the target model.

After retraining, validation performance provides reward $G_t$, which updates the policy $\pi_\theta$ and critic $R_\phi$. Thus, the framework does not select examples by a fixed reward threshold; instead, it learns across iterations which recurring failure modes are most useful for improving robustness.

Although the example is shown for NLI, the same failure filtering, automated validation, failure-mode clustering, and policy-guided sampling mechanism can be applied to other classification and reasoning tasks considered in this work.

### B.13 Sensitivity to Selection Threshold and Reward Noise

We analyze the robustness of our framework with respect to the selection threshold $\tau$ and noise in reward estimation. Since $\tau$ controls the trade-off between data quality and coverage, and reward estimates are derived from noisy downstream feedback, understanding their impact is critical for stable optimization.

**Sensitivity to Selection Threshold.** We varied $\tau$ over a wide range relative to the empirical reward distribution, selecting values corresponding to the 60th, 70th, 80th, and 90th reward percentiles. Lower thresholds admit more adversarial candidates, while higher thresholds enforce stricter filtering. All experiments were conducted using $\alpha = 0.83$ and a 1:4 mixing ratio.

**Reward Noise Injection.** To simulate imperfect reward estimation, we injected additive Gaussian noise into the predicted reward:

$$\tilde{r}(x) = r(x) + \epsilon, \quad \epsilon \sim \mathcal{N}(0, \sigma^2), \tag{84}$$

where $\sigma$ controls noise magnitude. We evaluate $\sigma \in \{0.05, 0.1, 0.2\}$, covering mild to severe corruption regimes.

Table 10 reports performance under varying $\tau$ and noise levels on SNLI. Performance remains stable across a broad operating region. Moderate deviations from the default threshold ($\tau^\star$) incur only minor degradation, and the system tolerates substantial reward noise before significant accuracy loss occurs.

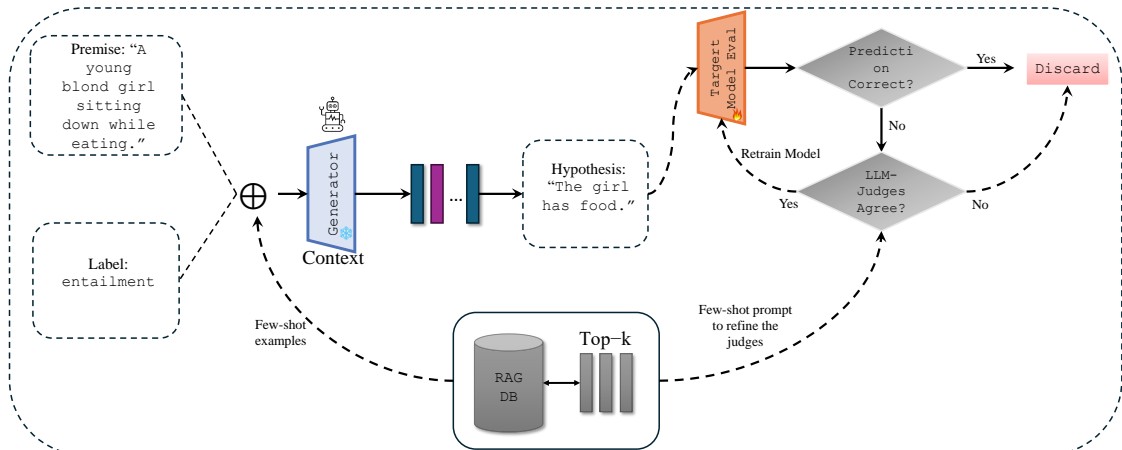

Figure 12: Illustration of one failure-mode contextual bandit curation iteration on a Natural Language Inference (NLI) example. Given a premise and label, retrieval-augmented prompting generates candidate hypotheses, which are first filtered by the target model to retain incorrect predictions and then validated by an automated LLM judge ensemble. Validated failures are clustered into failure modes, and a contextual-bandit policy selects which modes to sample for retraining under an adversarial budget. Validation reward $G_t$ updates the policy $\pi_\theta$ and critic $R_\phi$, enabling adaptive selection of high-impact failure modes across iterations.

Table 10: Sensitivity to selection threshold $\tau$ and reward noise (SNLI, 1:4 gen:orig, $\alpha = 0.83$).

| $\tau$ (Percentile) | Noise $\sigma$ | Accuracy (%) |
|---|---|---|
| 60% | 0.00 | 91.94 |
| 70% | 0.00 | 92.21 |
| 80% ($\tau^\star$) | 0.00 | 92.60 |
| 90% | 0.00 | 92.17 |
| 80% | 0.05 | 92.41 |
| 80% | 0.10 | 92.05 |
| 80% | 0.20 | 91.62 |

Results indicate that the proposed framework operates in a broad stability regime. The performance plateau around $\tau^\star$ suggests that the system is not finely tuned to a narrow threshold range. Moreover, robustness to moderate reward noise is consistent with our theoretical bounded-drift analysis (Section 3.4), which guarantees controlled distributional evolution under noisy feedback. Together, these findings demonstrate that our approach is resilient to practical imperfections in reward estimation and threshold calibration.

### B.14 Example - Few-Shot Chat Sequence

**BGE based retrieval**   The chat sequences below present a clear few-shot retrieval sequence for a natural language inference task. They illustrate six premise-hypothesis pairs-two each for entailment, neutral, and contradiction-and conclude with a concise model prompt. This format makes the example selection process transparent and highlights the model's reasoning in a single, easily readable block. These examples are based solely on BGE retrieval.

---

**Example 1: Few-Shot Retrieval & Model Return**

**Shot 1**
**Premise**: A blond little girl enjoying a burrito.
**Label**: entailment.
**Hypothesis:** The girl ate a burrito.
**Shot 2**
**Premise**: A young blond girl sitting down while eating.
**Label**: entailment.
**Hypothesis:** The girl has food.
**Shot 3**
**Premise**: A blond little girl enjoying a burrito.
**Label**: neutral.
**Hypothesis:** The hungry girl ate a burrito at the restaurant.
**Shot 4**
**Premise**: A young blond girl sitting down while eating.
**Label**: neutral.
**Hypothesis:** The girl is eating at a picnic.
**Shot 5**
**Premise**: A blond little girl enjoying a burrito.
**Label**: contradiction.
**Hypothesis:** The brunette girl didn't like the burrito.
**Shot 6**
**Premise**: A young blond girl sitting down while eating.
**Label**: contradiction.
**Hypothesis:** The girl runs all over her house while eating because she can never sit down.
**Llama Generation**
**User:** Now generate a one-sentence hypothesis that contradicts the premise above. Return only the hypothesis without narration.
**Assistant (Llama):** The girl threw the sandwich away.

---

**Example 2: Few-Shot Retrieval & Model Return**

**Shot 1**
**Premise**: An older man is drinking orange juice at a restaurant.
**Label**: entailment.
**Hypothesis:** A man is drinking juice.
**Shot 2**
**Premise**: An older man is drinking orange juice at a restaurant.
**Label**: entailment.
**Hypothesis:** A man is drinking juice.
**Shot 3**
**Premise**: An older man sits with his orange juice at a small table in a coffee shop while employees in bright colored shirts smile in the background.
**Label**: neutral.
**Hypothesis:** An elderly man sits in a small shop.
**Shot 4**
**Premise**: An older man sits with his orange juice at a small table in a coffee shop while employees in bright colored shirts smile in the background.
**Label**: neutral.
**Hypothesis:** An older man drinks his juice as he waits for his daughter to get off work.
**Shot 5**
**Premise**: An older man sits with his orange juice at a small table in a coffee shop while employees in bright colored shirts smile in the background.
**Label**: contradiction.
**Hypothesis:** A boy flips a burger.
**Shot 6**
**Premise**: An older man is drinking orange juice at a restaurant.
**Label**: contradiction.
**Hypothesis:** Two women are at a restaurant drinking wine.
**Llama Generation**
**User:** Now generate a one-sentence hypothesis that contradicts the premise above. Return only the hypothesis without narration.
**Assistant (Llama):** The coffee shop is empty.

**Example 3: Few-Shot Retrieval & Model Return**

**Shot 1**
**Premise**: A few people in a restaurant setting, one of them is drinking orange juice.
**Label**: entailment.
**Hypothesis:** The diners are at a restaurant.
**Shot 2**
**Premise**: A few people in a restaurant setting, one of them is drinking orange juice.
**Label**: entailment.
**Hypothesis:** The diners are at a restaurant.
**Shot 3**
**Premise**: A few people in a restaurant setting, one of them is drinking orange juice.
**Label**: neutral.
**Hypothesis:** The people are eating omelettes.
**Shot 4**
**Premise**: A few people in a restaurant setting, one of them is drinking orange juice.
**Label**: neutral.
**Hypothesis:** The people are eating omelettes.
**Shot 5**
**Premise**: A few people in a restaurant setting, one of them is drinking orange juice.
**Label**: contradiction.
**Hypothesis:** The people are sitting at desks in school.
**Shot 6**
**Premise**: A few people are observing the orange juice section at the grocery store.
**Label**: contradiction.
**Hypothesis:** The people are at a baseball stadium.
**Llama Generation**
**User:** Now generate a one-sentence hypothesis that contradicts the premise above. Return only the hypothesis without narration.
**Assistant (Llama):** The people are at a concert.

**Optimized BGE + BM25 Retrieval with Tuned Alpha**    The paragraph below describes an optimized retrieval approach that combines semantic search using BGE embeddings with lexical scoring via BM25. By introducing a tunable alpha parameter, the method linearly interpolates between embedding-based similarity and keyword matching-enabling a dynamic balance that maximizes relevance across diverse query types. Through cross-validation, the optimal alpha value was determined to achieve superior retrieval performance compared to either technique alone.

---

**Example 1: Few-Shot Retrieval & Model Return**

**Shot 1**
**Premise**: A person on a horse jumps over a broken down airplane.
**Label**: entailment.
**Hypothesis:** A person is outdoors, on a horse.
**Shot 2**
**Premise**: A person on a horse jumps over a broken down airplane.
**Label**: entailment.
**Hypothesis:** A person is outdoors, on a horse.
**Shot 3**
**Premise**: A person on a horse jumps over a broken down airplane.
**Label**: neutral.
**Hypothesis:** A person is training his horse for a competition.
**Shot 4**
**Premise**: A person is riding a horse, which is jumping over a grounded airplane.
**Label**: neutral.
**Hypothesis:** The horse jumps over a toy airplane.
**Shot 5**
**Premise**: A person on a horse jumps over a broken down airplane.
**Label**: contradiction.
**Hypothesis:** A person is at a diner, ordering an omelette.
**Shot 6**
**Premise**: A person on a horse jumps over a broken down airplane.
**Label**: contradiction.
**Hypothesis:** A person is at a diner, ordering an omelette.
**Llama Generation**
**User:** Now generate a one-sentence hypothesis that is neutral with the premise above. Return only the hypothesis without narration.
**Assistant (Llama):** The horse is very well trained.

**Example 2: Few-Shot Retrieval & Model Return**

**Shot 1**
**Premise**: Children smiling and waving at camera.
**Label**: entailment.
**Hypothesis:** There are children present.
**Shot 2**
**Premise**: Children smiling and waving at camera.
**Label**: entailment.
**Hypothesis:** There are children present.
**Shot 3**
**Premise**: Children smiling and waving at camera.
**Label**: neutral.
**Hypothesis:** They are smiling at their parents.
**Shot 4**
**Premise**: Children smiling and waving at camera.
**Label**: neutral.
**Hypothesis:** They are smiling at their parents.
**Shot 5**
**Premise**: Children smiling and waving at camera.
**Label**: contradiction.
**Hypothesis:** The kids are frowning.
**Shot 6**
**Premise**: Children smiling and making faces to the camera.
**Label**: contradiction.
**Hypothesis:** The children are deceased.
**Llama Generation**
**User:** Now generate a one-sentence hypothesis that contradicts the premise above. Return only the hypothesis without narration.
**Assistant (Llama):** The children are asleep.

**Example 3: Few-Shot Retrieval & Model Return**

**Shot 1**
**Premise**: An older man is drinking orange juice at a restaurant.
**Label**: entailment.
**Hypothesis:** A man is drinking juice.
**Shot 2**
**Premise**: An older man is drinking orange juice at a restaurant.
**Label**: entailment.
**Hypothesis:** A man is drinking juice.
**Shot 3**
**Premise**: An older man is drinking orange juice at a restaurant.
**Label**: neutral.
**Hypothesis:** A man in a restaurant is waiting for his meal to arrive.
**Shot 4**
**Premise**: An elderly man is drinking orange juice at a cafe.
**Label**: neutral.
**Hypothesis:** An older gentleman is enjoying his orange juice at a new cafe.
**Shot 5**
**Premise**: An older man is drinking orange juice at a restaurant.
**Label**: contradiction.
**Hypothesis:** Two women are at a restaurant drinking wine.
**Shot 6**
**Premise**: An older man is drinking orange juice at a restaurant.
**Label**: contradiction.
**Hypothesis:** Two women are at a restaurant drinking wine.
**Llama Generation**
**User:** Now generate a one-sentence hypothesis that is neutral with the premise above. Return only the hypothesis without narration.
**Assistant (Llama):** The restaurant has a lot of tables.

**BM25 based retrieval**   The retrieval process relies exclusively on the BM25 algorithm, which computes scores based on term frequency, document frequency, and document length normalization. By focusing on exact keyword matches and weighting rarer terms more heavily, BM25 effectively ranks documents that share the most relevant vocabulary with the query. This approach is well-suited for tasks where precise lexical overlap is paramount and semantic generalization is less critical. However, it may miss contextually related content when synonyms or paraphrases are used.

---

**Example 1: Few-Shot Retrieval & Model Return**

**Shot 1**
**Premise**: An older man is drinking orange juice at a restaurant.
**Label**: entailment.
**Hypothesis:** A man is drinking juice.
**Shot 2**
**Premise**: An elderly man is drinking orange juice at a cafe.
**Label**: entailment.
**Hypothesis:** An old man is enjoying a beverage at a cafe.
**Shot 3**
**Premise**: An older man is drinking orange juice at a restaurant.
**Label**: neutral.
**Hypothesis:** A man in a restaurant is waiting for his meal to arrive.
**Shot 4**
**Premise**: An older man is drinking orange juice at a restaurant.
**Label**: neutral.
**Hypothesis:** A man in a restaurant is waiting for his meal to arrive.
**Shot 5**
**Premise**: An older man is drinking orange juice at a restaurant.
**Label**: contradiction.
**Hypothesis:** Two women are at a restaurant drinking wine.
**Shot 6**
**Premise**: An older man is drinking orange juice at a restaurant.
**Label**: contradiction.
**Hypothesis:** Two women are at a restaurant drinking wine.
**Llama Generation**
**User:** Now generate a one-sentence hypothesis that entails the premise above. Return only the hypothesis without narration.
**Assistant (Llama):** A man is drinking orange juice at a restaurant.

**Example 2: Few-Shot Retrieval & Model Return**

**Shot 1**
**Premise**: An older man is drinking orange juice at a restaurant.
**Label**: entailment.
**Hypothesis:** A man is drinking juice.
**Shot 2**
**Premise**: An older man is drinking orange juice at a restaurant.
**Label**: entailment.
**Hypothesis:** A man is drinking juice.
**Shot 3**
**Premise**: An older man is drinking orange juice at a restaurant.
**Label**: neutral.
**Hypothesis:** A man in a restaurant is waiting for his meal to arrive.
**Shot 4**
**Premise**: An older man is drinking orange juice at a restaurant.
**Label**: neutral.
**Hypothesis:** A man in a restaurant is waiting for his meal to arrive.
**Shot 5**
**Premise**: An older man is drinking orange juice at a restaurant.
**Label**: contradiction.
**Hypothesis:** Two women are at a restaurant drinking wine.
**Shot 6**
**Premise**: An elderly man is drinking orange juice at a cafe.
**Label**: contradiction.
**Hypothesis:** An elderly man is drinking apple juice at a bar.
**Llama Generation**
**User:** Now generate a one-sentence hypothesis that contradicts the premise above. Return only the hypothesis without narration.
**Assistant (Llama):** The man is drinking coffee.

---

**Example 3: Few-Shot Retrieval & Model Return**

**Shot 1**
**Premise**: A woman in a green jacket and hood over her head looking towards a valley.
**Label**: entailment.
**Hypothesis:** The woman is wearing green.
**Shot 2**
**Premise**: A woman in a green jacket and hood over her head looking towards a valley.
**Label**: entailment.
**Hypothesis:** The woman is wearing green.
**Shot 3**
**Premise**: A woman in a green jacket and hood over her head looking towards a valley.
**Label**: neutral.
**Hypothesis:** The woman is cold.
**Shot 4**
**Premise**: A woman gazes over the valley below.
**Label**: neutral.
**Hypothesis:** she looks at the valley she owns.
**Shot 5**
**Premise**: A woman in a green jacket and hood over her head looking towards a valley.
**Label**: contradiction.
**Hypothesis:** The woman is nake.
**Shot 6**
**Premise**: A woman in a green jacket and hood over her head looking towards a valley.
**Label**: contradiction.
**Hypothesis:** The woman is nake.
**Llama Generation**
**User:** Now generate a one-sentence hypothesis that is neutral with the premise above. Return only the hypothesis without narration.
**Assistant (Llama):** The woman is looking at something.

## B.15 Prompt Design for Task-Specific Candidate Generation

We employ task-specific prompting strategies to guide large language models in generating adversarial candidates consistent with the target supervision signal. Prompts are designed to be concise, label-conditioned, and deterministic, ensuring controllable hypothesis synthesis and high semantic fidelity. All prompts instruct the model to return only the generated output without additional narration.

### B.15.1 SNLI Prompting Strategy

For the Natural Language Inference task, the goal is to generate a single-sentence hypothesis whose semantic relation to the given premise matches a specified target label $y \in \{entailment, neutral, contradiction\}$. Given a premise $p$ and target label $y$, we use the following template:

> **System:** You are a language expert that helps create an NLI dataset. Given a premise sentence and a desired label, your job is to provide a one-sentence hypothesis, such that the label is relevant to the relation between the given premise and your generated hypothesis. Make sure to keep the hypothesis short and no longer than a sentence.
>
> **User:**
> Premise: {premise}
>
> Desired label: {label}
>
> Now generate a one sentence hypothesis that {relation} the premise above. Return only the hypothesis without narration.

Here, `{label}` corresponds to the target class (entailment, neutral, contradiction), and `{relation}` maps to the appropriate semantic relation ("entails", "is neutral with", "contradicts"). The prompt enforces minimal length and discourages explanatory text.

We further employ low-temperature decoding to reduce sampling variance and ensure consistent adversarial patterns across iterations.

### B.15.2 FEVER Prompting Strategy

For the FEVER fact verification task, the objective is to generate evidence claims whose veracity can be evaluated with respect to a given document or knowledge source. Given an evidence context $e$ and a target label $y \in \{\text{SUPPORTS}, \text{REFUTES}, \text{NOT\_ENOUGH\_INFO}\}$, we use the following template:

> **System:** You are a language expert that helps create fact verification datasets. Given evidence text and a desired label, your job is to generate a single-sentence claim that matches the specified verification outcome. Make sure the claim is concise and factual.
>
> **User:**
> Evidence: {evidence}
>
> Desired label: {label}
>
> Now generate a one sentence claim that is {relation} by the evidence above. Return only the claim without narration.

Here, `{label}` corresponds to the FEVER classes, and `{relation}` maps to "supported by", "refuted by", or "cannot be verified from". This formulation encourages the model to synthesize claims that are directly grounded in the provided evidence.

As in the NLI setting, we constrain generation length and apply low-temperature sampling to prioritize precision over diversity.

### B.15.3 Design Rationale

Across tasks, prompt templates are designed to satisfy three principles: (i) explicit conditioning on the target label, (ii) minimal linguistic ambiguity, and (iii) strict output formatting. This enables stable generation, reliable automated validation, and consistent reward estimation, facilitating effective failure-aware adversarial data curation.

