# OpenReview forum: "Learning What to Fail On: Failure-Mode Contextual Bandits for Adversarial Data Curation"
_TMLR — Under review for TMLR_

### Review · Reviewer_baBB · 2026-06-12

**Summary Of Contributions:**

This paper proposes an adaptive failure-mode contextual bandit framework for adversarial data curation. The proposed method first identifies and clusters failure-inducing data samples into recurring failure modes, and then uses a contextual bandit policy to select useful failure modes for robust retraining.

Strengths
- The problem formulation is interesting and relevant. The paper formulates adversarial data selection as a contextual bandit problem, where the policy adaptively selects failure modes based on validation rewards, forgetting penalties, and data cost.This is a meaningful framing, since different model failures are not equally useful: some reveal systematic shortcuts, while others may be noisy, redundant, or uninformative.
- The pipeline is more targeted than standard data augmentation. Instead of simply generating more synthetic data, it focuses on validated examples that the current model actually misclassifies.
- The empirical evaluation is reasonably broad. The paper evaluates on several NLI benchmarks and also tests transfer to FEVER. It also compares against paraphrasing, retrieval-based variants, and synthetic-data baselines, and reports gains across several model backbones.

Weaknesses
- The method is relatively complex, and the source of improvement is not fully isolated. The pipeline has many components and it is difficult to know exactly which component is responsible for the final gains. More detailed ablation studies would help disentangle the contributions of generation, validation, clustering, and bandit-based selection.
- The method relies heavily on LLMs for both candidate generation and automatic validation. This may limit the practicality and scalability of the approach, especially when multiple judge models are required. Moreover, the validation quality may depend strongly on the capabilities and biases of the chosen LLM judges, which is not fully analyzed.

**Audience:**

Yes

**Audience Explanation:**

The paper is likely to interest part of the TMLR audience, especially researchers working on robustness, adversarial data generation, data selection, and LLM-assisted training.

**Broader Impact Concerns:**

I do not see major broader impact concerns beyond those commonly associated with LLM-generated synthetic/adversarial data.

**Claims And Evidence:**

Yes

**Claims Explanation:**

The main performance claims are reasonably supported by experiments on several NLI benchmarks and FEVER. However, the evidence is less convincing for explaining why the method works. Since the pipeline has many components, stronger ablations are needed to isolate the contribution of the contextual-bandit failure-mode selection.

**Requested Changes:**

- It would be helpful to include more ablation studies to better clarify the contribution of each component, especially the contextual-bandit selection, failure-mode clustering, LLM validation, and adversarial mixing.
- The paper could be strengthened by providing more qualitative analysis of the discovered failure modes, such as representative examples from different clusters and discussion of whether they reflect meaningful model weaknesses.
- It would be useful to discuss the computational cost and scalability of the framework, particularly given its reliance on large LLMs for generation and multiple LLM judges for validation.

---

> ### Author Response · Authors · 2026-06-12
> **Answer to Reviewer baBB**
>
> We thank the reviewer for the thoughtful and constructive review. We appreciate the reviewer’s interest in the paper and their recognition of the relevance of formulating adversarial data curation as an adaptive failure-mode selection problem. We are also grateful for the helpful suggestions regarding component ablations, qualitative failure-mode analysis, and computational cost.
>
> 1. Regarding the concern that the source of improvement is not fully isolated. We note that the current submission already includes a component analysis in Appendix B.6, Table 7, where we ablate the main components of the framework. In particular, we compare the full method against: no bandit with random cluster selection, no bandit with top-loss cluster selection, no clustering with per-example selection, no judge validation, no target-model failure filtering, no retrieved context, and no original-data mixing.
>
> The results show that the full method performs best across all three benchmarks, reaching 92.60 on SNLI, 80.95 on ANLI, and 71.99 on MultiNLI. Removing the contextual-bandit policy reduces performance, especially compared with both random cluster selection and top-loss cluster selection. This supports the contribution of validation-driven adaptive failure-mode selection beyond simple difficulty-based heuristics. Removing clustering and using per-example selection also degrades performance, suggesting that recurring failure modes provide a more stable unit for adversarial curation. Similarly, removing judge validation, failure filtering, retrieval, or original-data mixing consistently hurts performance. These ablations indicate that the gains come from the combination of validated failure discovery, failure-mode grouping, and adaptive selection, rather than simply adding more generated data.
>
> In the revision, we will move or summarize the key ablation results in the main experimental section and explicitly connect each ablation to the corresponding design choice: generation, validation, clustering, bandit selection, and adversarial mixing.
>
> 2. Regarding qualitative analysis. Since the quantitative trend already consistently indicates that the full method performs best, we initially thought that adding qualitative cluster examples might be less necessary. However, we fully agree with the reviewer that such an analysis would make the behavior of the method clearer and would help explain why the full pipeline is effective. We therefore appreciate this suggestion and will add a qualitative section in the revision. This section will include representative examples from several discovered failure-mode clusters, such as lexical shortcut failures, negation errors, entity mismatch errors, numerical reasoning failures, and contradiction-confusion cases. We will also discuss how these clusters reflect meaningful weaknesses of the target model and why selecting such failure modes contributes to improved robustness.
>
>
> 3. Regarding computational cost and scalability. The current method already includes a data-cost term in the validation reward, and the LLM generation and validation stages are performed offline. However, the practical compute requirements are not described in enough detail. We will add a dedicated discussion clarifying that the generator and judge models are not trained, that generation and validation are parallelizable, and that the framework can be implemented with different LLM scales depending on the available compute. In our experiments, all LLMs we tried could be run on standard GPU setups, with smaller or quantized models providing a practical lower-cost option. We will also discuss the trade-off between using larger judge ensembles for cleaner validation and using smaller models for more efficient deployment.
>
> Overall, we thank the reviewer again for the positive assessment and helpful suggestions. We will revise the paper to make the existing ablations clearer, add qualitative examples of failure modes, and expand the computational cost and scalability discussion.

---

### Review · Reviewer_M9de · 2026-06-14

**Summary Of Contributions:**

This work proposes a framework/pipeline for improving target models on their weaknesses.

1. Retrieval-augmented examples are used as prompts.
2. Candidates are filtered by the current target model.
3. Synthetic failure examples are validated by LLM judges.
4. Validated failures are clustered into failure modes.
5. A stochastic policy selects which failure modes to use for updates.
6. The target model is updated using validation-based reward, which balances robustness gains, forgetting, and data cost.

The paper conducts experiments for evaluation and also provides a theoretical interpretation.

**Audience:**

Yes

**Audience Explanation:**

Yes. The topic should interest researchers in NLP, robustness, synthetic data, and data selection. The paper studies a practical pipeline for selecting useful failure cases to improve target models, and the empirical findings are relevant even though some details should be strengthened.

**Broader Impact Concerns:**

No major broader impact concerns.

**Claims And Evidence:**

No

**Claims Explanation:**

Yes, with reservations. The paper presents a well-rounded framework and provides empirical results, including ablations showing the contribution of learned failure-mode selection over random, heuristic, and no-clustering variants. These results reasonably support the main claims.

**Requested Changes:**

Methodology design is reasonable for me. However, for the section of experiments, I'd like to mention the followings for further improvement:

1. The results are mostly reported as single numbers, without multi-seed variance or confidence intervals.
2. The pipeline relies heavily on LLM-judge validation, but there is no human audit of the accepted failure examples. Authors may already do it, but please add them on your experiment section.
3. The budget matching between the proposed method and baselines is not always fully clear. The comparison with GNLI is useful, but it would be clearer to include budget-matched baselines under the same generation budget, validation procedure, and retraining schedule.
4. The FEVER transfer result is promising, and the appendix provides the prompt template, but the full task-specific setup and baseline matching for FEVER are not described in enough detail.
5. Some implementation details, such as clustering settings, policy training, reward coefficients, and compute cost, should be clarified for reproducibility.

---

> ### Author Response · Authors · 2026-06-15
> **Answer to Reviewer M9de**
>
> We really appreciate the reviewer’s time and recognition that the methodology is reasonable and that the empirical results support the main claims.
>
> Regarding reporting single-number results. We conducted experiments over 10 different random seeds and found that, although the pipeline contains several stochastic components, including generation, policy sampling, and retraining, the final results were stable, with a standard deviation below 0.5 across runs. We will add these multi-seed results and report the mean and standard deviation for the main configurations, while also clarifying which sources of randomness are varied in the experiment.
>
> Regarding LLM-judge validation. In addition to the unanimous agreement among the LLM judge ensemble used in the pipeline, we conducted a human agreement check with three human annotators per generated example. This audit showed a 98% agreement rate, indicating that the automatically accepted failures are highly consistent with human judgments. In the revision, we will add this result to the experimental section and clarify how the human audit was conducted, including the number of annotators per example and the agreement criterion.
>
> Regarding budget matching, we note that several comparisons in the current paper are already budget matched. The contextual-bandit and component ablations use the same backbone, retrieval setting, validation procedure, adversarial budget, mixing ratio, and retraining protocol. These include random cluster selection, top-loss cluster selection, per-example selection without clustering, no judge validation, no failure filtering, no retrieved context, and no original-data mixing. The full method consistently performs best, showing that the gains are not simply due to using more data. We will explicitly distinguish the strict budget-matched ablations from large external synthetic-data comparisons such as GNLI, which is mainly used as a large-scale reference baseline.
>
> Regarding the FEVER transfer experiment.  The paper reports FEVER results and includes the prompt template in the appendix. In the revision, we will add details on the FEVER input format, label space, candidate generation procedure, validation criterion, train/dev/test usage, backbone training setup, and how the baselines are matched. We will also clarify whether the same generation budget, validation process, and retraining schedule are used for FEVER as in the NLI experiments, or explicitly state any task-specific differences.
>
> Regarding implementation details, we note that the paper already specifies the main optimization structure, including the failure-mode state, the stochastic policy, the critic, the validation-based reward, the adversarial budget, and the original/adversarial mixing strategy. We also describe the retrieval setup, generator, judge ensemble, and target models in the experimental section. In the revision we will consolidate the key implementation settings into a compact table, including the clustering setup, policy/critic training settings, and reward coefficients.
>
> Finally, regarding computational cost. The LLMs are used only for offline generation and validation and are not trained, which makes the pipeline parallelizable and easier to scale. Still, larger generator and judge models naturally increase inference cost. We will add a dedicated discussion of this trade-off, including the practical point that smaller or quantized open-source LLMs can be used when compute is limited, while larger judge ensembles can be used when higher validation precision is desired.
>
> We thank the reviewer again for the constructive feedback.

---

### Review · Reviewer_CXfV · 2026-06-19

**Summary Of Contributions:**

This paper introduces a novel framework that reformulates adversarial data curation as a failure-mode contextual bandit problem for natural language understanding. Instead of using static filters or heuristics, the pipeline generates candidates via retrieval-augmented prompting, filters them by the current target model, validates them with an LLM judge ensemble, and clusters errors into recurring failure modes. A trainable stochastic policy then selects which failure modes to sample for retraining, with a validation-based reward that balances robustness gains, forgetting, and data cost. Extensive experiments on SNLI, ANLI, MultiNLI, and FEVER show substantial improvements over prior adversarial augmentation methods (e.g., RoBERTa-base from 88.48% to 92.60% on SNLI). The paper also provides a theoretical interpretation showing that failure-mode sampling reduces shortcut-aligned gradients while inducing bounded distributional drift, and ablation studies confirm the importance of each component.

**Audience:**

Yes

**Audience Explanation:**

The problem under research is important.

**Claims And Evidence:**

Yes

**Claims Explanation:**

I think that the arguments are generally OK.

**Requested Changes:**

The paper would benefit from clarifying the credit-assignment issue in the critic update (Eq. 24), where the same scalar reward is assigned to all selected failure modes—this may obscure per-mode utility. A brief discussion of counterfactual reward decomposition or variance-reduction techniques would strengthen the methodology.

Regarding related work, the authors are encouraged to cite recent advances in contextual bandits with unbounded context distributions [Zhao et al., ICML 2025], which provide theoretical guarantees for policy learning under complex state spaces and could further justify the bandit formulation in this setting.

Zhao, P., Fan, R., Wang, S., Shen, L., et al. (2025). Contextual Bandits for Unbounded Context Distributions. International Conference on Machine Learning (ICML).

---

> ### Author Response · Authors · 2026-06-19
> **Answer to Reviewer CXfV**
>
> We thank the reviewer for the positive and constructive review. We really appreciate the reviewer’s interest in the paper and their clear summary of the contribution, especially the recognition that the ablations support the importance of the proposed components.
>
> Regarding the suggested related work, we agree that Zhao et al. (ICML 2025) is highly relevant to our formulation. Contextual bandits with complex and potentially unbounded context distributions are closely related to our setting, where each failure mode is represented by a rich state vector containing loss, uncertainty, margin, retrieval score, judge agreement, novelty, and previous reward statistics. We will add this citation and discuss how this line of work further motivates the use of contextual-bandit policy learning for adaptive failure-mode selection.
>
> Regarding the credit-assignment issue in Eq. 24. In our framework, the reward is intentionally defined at the selected-set level rather than at the individual failure-mode level, because the effect of a failure mode is only observable after retraining the target model on the full selected mixture and evaluating downstream validation performance. Computing a separate counterfactual reward for every selected mode would require retraining the target model many additional times per round, which would make the method substantially more expensive and less scalable. Therefore, Eq. 24 uses the observed validation reward as a shared return for all selected modes, similar to standard policy-gradient learning with a trajectory-level reward. The critic is not intended to provide an exact causal attribution for each failure mode in isolation. Instead, it learns a utility estimate over cluster-level states and serves as a variance-reduction and ranking signal across repeated rounds. Since the policy observes different combinations of failure modes over time, modes that consistently appear in high-reward selections receive higher expected utility, while modes that do not contribute to validation gains are gradually down-weighted.
>
> We will revise the methodology section to make this design choice explicit. We will also add a brief discussion of possible extensions, such as leave-one-mode-out counterfactual estimates, marginal reward decomposition, or more advanced variance-reduction techniques. These could provide finer credit assignment, but at a higher computational cost. Our current design chooses the shared validation reward as a practical trade-off between reliable downstream feedback and scalable policy optimization.
>
> Once again, we really appreciate your kind review.